# Fluxes of gaseous elemental mercury (GEM) in the High Arctic during atmospheric mercury depletion events (AMDEs)

Jesper Kamp[1,2], Henrik Skov[1,3], Bjarne Jensen[1,3] and Lise Lotte Sørensen[1,3]

[1]Arctic Research Centre, Aarhus University, 8000 Aarhus, Denmark
[2]Department of Engineering, Aarhus University, 8000 Aarhus, Denmark
[3]Department of Environmental Science, Aarhus University, 4000 Roskilde, Denmark

*Correspondence to*: Jesper Kamp (JK@eng.au.dk)

**Abstract.** Measurements of gaseous elemental mercury (GEM) fluxes over snow surfaces using a relaxed eddy accumulation (REA) system are carried out at the High Arctic site Villum Research Station, Station Nord in North Greenland. Simultaneously, $CO_2$ fluxes are determined using the eddy covariance (EC) technique. The REA system with dual-inlets and dual-analyzers are used to measure fluxes directly over the snow. The measurements were carried out from April 23 to May 12 during spring 2016, where atmospheric mercury depletion events (AMDEs) took place. The measurements showed a net emission of 8.9 ng $m^{-2}$ $min^{-1}$, with only a few minor episodes of net depositional fluxes, from a maximum deposition of 8.1 ng $m^{-2}$ $min^{-1}$ to a maximum emission of 179.2 ng $m^{-2}$ $min^{-1}$. The data support the theory that gaseous oxidized mercury (GOM) is deposited during AMDEs followed by formation of GEM on surface snow and is reemitted as GEM shortly after the AMDEs. Furthermore, observation of the relation between GEM fluxes and atmospheric temperature suggest that GEM emission partly could be affected by surface heating. However, it is also clear that the GEM emissions are affected by many parameters.

## 1 Introduction

Mercury (Hg) is a toxic element found in the atmosphere primarily as elemental mercury. Airborne Hg can have several forms: gaseous oxidized mercury (GOM), particulate bound mercury (PBM) or gaseous elemental mercury (GEM). PBM and GOM are removed faster from the atmosphere than GEM and have atmospheric lifetimes in the order of days (Sørensen et al., 2010;Goodsite et al., 2004, 2012;Valente et al., 2007). Thus, GOM and PBM generally deposit near emission sources. The lifetime of GEM, determined by the reaction between GEM and Br (Goodsite et al., 2012, 2004), spans from one to two months (Holmes et al., 2006;Sørensen et al., 2010). Thus, GEM can be transported over longer distances to areas with low natural and anthropogenic emissions. GEM concentrations in the Arctic are mainly due to long-range transportation from lower latitude sources (Dastoor et al., 2008;Pandey et al., 2011;Christensen et al., 2004).

In the Arctic, sub-Arctic and Antarctic atmospheric mercury depletion events (AMDEs) have been observed in coastal areas during spring (Steffen et al., 2008;Dastoor et al., 2008) causing significant Hg deposition in Polar Regions (Steffen et al., 2008;Dastoor et al., 2008). During AMDEs, GEM is depleted from the atmosphere by oxidation to GOM (Skov et al.,

2004;Toyota et al., 2014), which is then deposited locally due to fast deposition limited only by aerodynamic resistance (Skov et al., 2006). Mercury bio-accumulates in Arctic marine wildlife through the food web; this is a human health concern in Arctic communities due to high mercury exposure through the traditional indigenous diet (AMAP, 2011).

Typical Arctic spring conditions such as low temperatures, sunlight and reactive halogens favor AMDEs (Brooks et al., 2006;Steffen et al., 2015;Goodsite et al., 2004, 2012;Berg et al., 2003). In earlier studies (Skov et al., 2004;Schroeder et al., 1998), depletion of ozone during AMDEs revealed a correlation between ozone and GEM concentrations. Ozone concentration decreases due to reaction with bromine: $O_3 + Br \rightarrow O_2 + BrO$ (Hausmann and Platt, 1994). Data from Villum Research Station (VRS), Station Nord in North Greenland suggests a common reactant responsible for the removal of GEM and ozone that agreed with Br reactions during AMDEs (Skov et al., 2004).

Following AMDEs, elevated concentrations of GEM have been observed (Lalonde et al., 2002;Steffen et al., 2008), and it is suggested that photochemical processes in the snow reduce deposited Hg back to GEM, which is then reemitted into the atmosphere (Ferrari et al., 2004;Lalonde et al., 2002). The reduction to GEM is assumed to take place in the aqueous phase and potentially in particles with significant water content (Steffen et al., 2015).

Knowledge of the dynamics of Hg in snow during AMDEs is important in order to understand the fate of GEM. Studies of Hg in snow evince an increase from February and peak in May (Steffen et al., 2014), likely due to the accumulation of deposited GOM, and this finding corresponds well with the peak occurrence of AMDEs in April and May (Steffen et al., 2015). A number of specific conditions and parameters, such as temperature, radiation and chemical composition of the snow affect the dynamics of Hg in the snowpack (Lalonde et al., 2002), but Hg in snow is mainly found in oxidized forms (Steffen et al., 2008).

The dynamics of Hg in snowpack have been studied previously, e.g. by Faïn et al. (2013) who observed complex GEM variations at a mid-latitude site in Colorado, USA. They found that GEM concentration in the top layers of the snowpack increased with increasing solar radiation, suggesting GEM production in the snowpack (Faïn et al., 2013) and that GEM production follows AMDE (Brooks et al., 2006). This is most likely due to photoreduction of GOM and subsequent emission of GEM; however, it is also possible that a correlation between solar radiation-induced parameters such as heat flux or temperature change and GEM fluxes exists, making it relevant to look into temperature and heat flux as well as radiation in relation to GEM flux.

A recent non-Arctic study with a similar setup to measure GEM flux during snowmelt in Degerö, Sweden revealed diurnal variations of fluxes showing deposition from midnight to noon and emissions from noon to midnight with a mean of $3.0 \pm 3.8$ ng m$^{-2}$ h$^{-1}$ (Osterwalder et al., 2016). Furthermore, Osterwalder et al. (2016) found significant difference between GEM fluxes during unstable, stable, and neutral conditions with a near-zero flux during stable conditions, emission during unstable conditions and deposition during neutral conditions.

Previous GEM flux studies in the Arctic were mainly performed using chamber methods (e.g. Ferrari et al. (2008)) and the aerodynamic gradient method (AGM) (e.g. (Brooks et al., 2006;Cobbett et al., 2007)). The overview in Table 1 clearly shows the large variations in GEM fluxes found by studies performed in the Arctic. Chamber methods are attractive methods for measuring fluxes because of their low cost and simplicity but they suffer from a number of weaknesses. They only capture the

flux over a small area, the chamber affects the surface over which the measurement is taken and they can modify physical properties such as light and temperature (Bowling et al., 1998;Fowler et al., 2001). This implies that the measured flux will differ from the natural flux. The AGM is not altering the surface; however, it requires a homogeneous surface several hundred meters upstream from the measurement site. Furthermore, it is assumed that the vertical profile is only a consequence of the

vertical turbulent transport; nevertheless fast chemical reactions can affect the profile. Strong stratification violates the assumption of gradient measurements, thus the relaxed eddy accumulation (REA) method is in our opinion the best possible option to measure GEM flux. The most direct flux measurement technique is the eddy covariance (EC) technique (Buzorius et al., 1998) but close to the surface this technique only works for fast responding monitors (sampling frequency >5 Hz), which is not available for Hg. Therefore, we chose to employ the REA method (Businger and Oncley, 1990) which is based on EC

and the method does not affect the surface. Oncley et al. (1993) reported results with agreement within 20% for EC and REA and a study by Hensen et al. (1996) shows agreement between EC and REA within 10%, a difference that is reported not to be significant because the main error for REA is the determination of the concentration difference.

The aim of the study presented here is to enhance the understanding of the processes controlling the fluxes of GEM over snow-covered surfaces during the Arctic spring, where AMDEs take place. The REA method (Businger and Oncley, 1990) is used

for the flux measurement in a setup with a dual inlet (Cobos et al., 2002;Osterwalder et al., 2016) and dual detectors. GEM fluxes have been determined with REA previously over agricultural soil (Cobos et al., 2002), in a winter wheat cropland (Sommar et al., 2013;Zhu et al., 2015a), in an urban environment, and in boreal peatland (Osterwalder et al., 2016), but never in the Arctic.

## 2 Materials and methods

### 2.1 Measurement site

From April 23 to May 12 of 2016, measurements of GEM flux, $CO_2$ flux, GEM concentration, wind speed, wind direction, atmospheric stability and temperature were carried out at "Flyger's hut", a part of Villum Research Station, Station Nord (VRS). The hut is located 2.5 km southeast of the central complex of the Danish military base Station Nord in North Greenland (81°36' N, 16°40' W) (Figure 1). The station is located in the world's largest national park (Rasch et al., 2015). Flyger's hut is

located at 81°34.90' N, 16°37.19' W southeast of Station Nord to minimize influence from local air pollution. The hut has been used as a monitoring site for the Arctic Monitoring and Assessment Programme (AMAP, 2011), since 1994. At this latitude, the polar day lasts from mid-April to September and the polar night lasts from mid-October until the end of February. The dominant wind directions measured locally are from the southwest, potentially with katabatic winds from the Greenlandic ice cap southwest of Flyger's hut. The wind distribution during the campaign is shown in Figure 2.

At the beginning of the measuring period at the end of April, the snow depth was 1.02-1.03 m. Little precipitation was observed and the snow depth varied between 0.94 m and 1.09 m during the campaign. When we ended the measurements, the depth was

1.00-1.03 m. The changes in snow depth are due to blowing snow or sublimation as the temperature never rose above -1.7℃, with a mean temperature of -16.7℃. Snowmelt did not remove the snow until mid-July.

## 2.2 Air mass trajectories

To evaluate the origin of the air masses, backward trajectories were calculated using the NOAA HYSPLIT model (Rolph et al., 2017;Stein et al., 2015). Trajectories are calculated every six hours as 24-hour backwards trajectories from a starting point at VRS at 20 meters above ground level. Four examples of trajectory plots of single trajectories and trajectory frequency are shown in Figures 3 and 4.

## 2.3 Local meteorological measurements

An ultrasonic anemometer (METEK, uSonic-3 Scientific), installed at 6.40 m above ground level, was used to measure the wind components in x-, y- and z-directions at 10 Hz (see Figure 5). Fifteen-minute averaged values were calculated for wind speed, wind direction, friction velocity, temperature, stability and turbulence intensity.

## 2.4 Measurement of GEM flux

Atmosphere-surface fluxes of GEM were measured using the REA technique proposed by Businger and Oncley (1990), where the vertical turbulent transported flux is estimated from:

$$F = b\ \sigma_w \left( \overline{C_{up}} - \overline{C_{down}} \right), \tag{1}$$

When applying the REA technique, slower responding sensors can be used, in contrast to the EC technique where faster responding sensors are required. In eq. (1), b is a proportionality factor (the Businger coefficient) which can be experimentally determined from sensible heat or another scalar flux; $\sigma_w$ is the standard deviation of the vertical wind speed; the overbar denotes a mean; and $C_{up}$ and $C_{down}$ are the true gas concentration in updrafts and downdrafts, respectively. Separation of updrafts ($C_{up}$) and downdrafts ($C_{down}$) is obtained by the sonic anemometer and fast shifting valves, which separates the airstream according to the direction of fluctuations in the vertical wind velocity.

The REA technique proposed by Businger and Oncley (1990) uses a constant flow rate accounted for by the addition of the Businger coefficient, discussed extensively elsewhere(e.g. (Gao, 1995;Gronholm et al., 2008;Tsai et al., 2012)). A constant value for b can be used, but it is preferable to determine b from site to site from other scalars like $CO_2$ or temperature under the assumption of scalar similarity (Gao, 1995).

Often a wind-controlled "deadband" is introduced to avoid sampling of eddies with a vertical velocity close to zero. A threshold above or below zero indicates this deadband, and the magnitude of the fluctuations of the vertical wind velocity must be larger than this threshold for air samples to be collected. This also decreases the switching frequency of the valves by removing many small fluctuations. As a consequence, the deadband will increase the concentration difference between updrafts and downdrafts, hence b is reduced to compensate for the increased difference (Ammann and Meixner, 2002).

The overall system is shown in Figure 5, the system consists of two automated Hg vapor analyzers (Tekran, model 2537X) used to measure the GEM concentrations in updrafts and downdrafts, respectively. Data from the two Hg analyzers was compiled on a PC inside Flyger's hut. The sampling inlets are located 5.69 m above ground. Osterwalder et al. (2016) and Zhu et al. (2015b) describe the advantages of using dual inlets, where temporally synchronous concentration determination of updrafts and downdrafts is the most obvious advantage. The Teflon tubes were heated to 50°C and each tube is connected to a three-way valve, which can either collect sample air or zero air. The zero air was delivered from a zero air generator in excess to the valves when not sampling. A CompactRIO processor (cRIO-9033, National Instruments) sets the position of the valves according to the vertical wind velocity measured with the ultrasonic anemometer. The software LabVIEW (National Instruments) was embedded on the CompactRIO processor with a real time module and a programmable FPGA for high-speed control directly in the hardware. This allowed control of valve positions and collection of data from the ultrasonic anemometer. The REA system was mounted on a boom on top of Flyger's hut. The boom was placed at the edge of the roof and directed towards the prevailing wind direction in order to minimize flow distortion from the hut.

The standard deviation of the vertical wind speed was obtained from previous wind measurements at Station Nord and used for selection of the deadband range to yield a robust b (Held et al., 2008;Ruppert et al., 2006). Thus, a fixed deadband of $\pm0.076$ m s$^{-1}$ is applied to all the data. Correction for dilution according to the opening times of the valves is performed according to (Sommar et al., 2013):

$$C_{up} = \frac{[c_{up} - c_{zero\ air}(1-\alpha_{up})]}{\alpha_{up}} \ and \ C_{down} = \frac{[c_{down} - c_{zero\ air}(1-\alpha_{down})]}{\alpha_{down}}, \qquad (2)$$

where $c_{up}$ and $c_{down}$ refers to the GEM concentration in updrafts ($c_{up}$) or downdrafts ($c_{down}$); $c_{zero\ air}$ is the GEM concentration in the zero air delivered to the valves. $\alpha_{up}$ and $\alpha_{down}$ refers to the fraction of time where the updrafts ($\alpha_{up}$) or downdrafts ($\alpha_{down}$) are collected. $C_{up}$ and $C_{down}$ are true corrected concentrations used in equation 1.

Tekran 2537 models are based on pre-concentration of Hg on gold cartridges followed by thermal desorption in a flow of inert argon gas, and Hg detection by Cold Vapor Atomic Fluorescence Spectrometry (CVAFS). UV light (253.7 nm) excites Hg atoms, which emit the absorbed energy by fluorescence. Collection on gold traps, thermal desorption and CVAFS is an accurate method to measure Hg content in the air. The detection limit is 0.1 ng m$^{-3}$ for the Tekran 2537 (Ma et al., 2015). The sampling interval is 15 minutes with a flow rate of 1.5 L min$^{-1}$ and auto calibration every 25 hours. Skov et al. (2004) estimate the reproducibility to be within 20% (95% confidence interval) for two Tekran mercury analyzers measuring above 0.5 ng m$^{-3}$.

## 2.5 CO$_2$ flux determination for calculation of b

We determine the proportionality factor b used to calculate fluxes of GEM from CO$_2$ fluxes assuming fluxes of all gases are transported by the turbulence in a similar way. In contrast to the GEM flux, CO$_2$ flux can be measured using the more direct EC method, thus b can be estimated from the measured CO$_2$ flux and CO$_2$ concentrations using Eq. 1.

Close to the REA flux system, an enclosed CO$_2$ gas analyzer (LI-7200, LI-COR Inc.) was mounted on the boom with the inlet directly below the ultrasonic anemometer 6.08 m above ground and above the GEM sample inlets. The gas analyzer measures

$CO_2$ and $H_2O$ concentration at 10 Hz to derive the EC flux of $CO_2$ and $H_2O$. The CompactRIO compiles all data from the gas analyzer, valve positions and meteorological data from the REA system. The flux of $CO_2$ was measured in order to determine b from the EC $CO_2$ flux and back-calculations of $CO_2$ concentration in updrafts and downdrafts compared to the valve positions (Gao, 1995;Ruppert et al., 2006). Similarly, b was determined from temperature flux measurements. For each interval, b is used to determine the REA flux of GEM.

Meteorological conditions or parameters, such as temperature, wind direction and speed, heat fluxes, relative humidity, pressure, and water vapor were measured for further analysis of the GEM fluxes. The Monin-Obukhov length (L) was calculated in order to estimate stability, as atmospheric stratification is expected to affect the surface exchange. Stability is often described as z/L, where z is the measurement height. In order to ensure data from a well-developed turbulent flow field and a reasonably constant wind direction, wind speeds below two m s$^{-1}$ were discarded.

For an ideal Gaussian joint probability distribution of the vertical wind speed and the scalar concentration, b has a well-defined value of 0.627 (Wyngaard and Moeng, 1992). However, experimentally determined b's for fluxes of heat, moisture and $CO_2$ typically range from 0.5 to 0.7 (e.g. Katul et al. (1996), Ammann and Meixner (2002), Sakabe et al. (2014)).

As mentioned a fixed deadband of 0.076 m s$^{-1}$ is introduced. Adding a deadband will affect the magnitude of b. In many applications, a dynamic deadband scaled with standard deviation of the vertical velocity w ($\sigma_w$) is used, which gives a smaller but relatively constant b (Hansen et al., 2013) according to Eq.3:

$$b = b_0 \exp\frac{-0.75 \cdot \omega_0}{\sigma_w} \qquad\qquad (3)$$

Where $b_0$ is b without the deadband and $\omega_0$ is the dynamic deadband. However, for practical reasons (limitation on processing time for data control and data collection) we used a fixed deadband causing b to vary with $\sigma_w$. The standard deviation of w measured in present study varied between 0.03 and 0.4 m s$^{-1}$. According to eq. 3, this will cause a variation of b (~ 0.2-0.8) depending on the size of $b_0$. Several researchers have studied the dependence of $b_0$ on the atmospheric dimensionless stability parameter z/L (z/L < 0 indicates unstable, z/L >0 stable and z/L= 0 neutral conditions). The majority of the studies (Andreas et al., 1998;Ammann and Meixner, 2002;Sakabe et al., 2014) showed an increase in $b_0$ with increasing z/L, however for the most part they refer to a limited stability range (-1.5< z/L <1.5). In the high Arctic, we often find very stable as well as neutral and slightly unstable stratification. In order to keep the estimated b values within a well-investigated stability range, data are discarded if they fall outside the stability range of -1.5< z/L <1.5. If b in a given experiment differs too much from the expected value, the probability distribution is likely to differ from the Gaussian distribution, thus in the present experiment, data was discarded in periods where b derived from T or $CO_2$ was below 0.2 and above 0.8.

After data filtration, 26% of the total 1653 measurements were approved during the campaign. We are aware that this is a very strict filtration; however, this ensures that the data used for the analysis are solid.

Several studies have been dedicated to investigate the implications on the flux related to b (e.g. Andreas et al. (1998), Ruppert et al. (2006) and Sakabe et al. (2014)) and the standard deviation of b is often estimated to be around 10% (e.g. Ammann and Meixner (2002), Sommar et al. (2013) and Sakabe et al. (2014)). However, b is calculated based on measurements of $CO_2$ fluxes, thus the uncertainty of b must be related to the uncertainty of the measured flux. It is not trivial to estimate the uncertainty of EC fluxes. Finkelstein and Sims (2001) suggested to use direct calculation of the variance of the covariance for calculating the random sampling error in EC measurements. They tested measurements at several types of surfaces and found the relative error to be approximately 25-30% for trace gas fluxes. However, one could argue that this method is only revealing how constant the flux measurement is and not how accurate the measured flux is. A more correct way to estimate the error is to measure the flux in parallel towers (Post et al., 2015). This is very expensive and very rarely carried out. Hence, here we use the general relative standard deviation of $CO_2$ fluxes on 25-30% estimated by Finkelstein and Sims (2001). Using error propagation theory on eq.1 the uncertainty of b ($u_b$) can be estimated as the combined relative uncertainty of the measured flux (25%) and the relative uncertainty of the measured concentration of $CO_2$ (1% (Li-Cor)) from following equation:

$$u_b(y) = \sqrt{\sum_{i=1}^{n} u(x_i)^2} \qquad (4)$$

Where $u(x_i)$ is the standard uncertainty. The uncertainty of b is $\approx 25\%$. To estimate the total uncertainty of the GEM flux we also have to consider the uncertainty of the measurements of the GEM concentration. This was found to be 10% by Skov et al. 2004, which used same type of instrument for GEM measurements. The uncertainty of the GEM flux can now be determined from the combined uncertainty of the concentration measurements and uncertainty of the estimated b: $\sqrt{0.1^2 + 0.1^2 + 0.25^2} \approx 0.30$ and the uncertainty of the flux becomes $\approx 60\%$ at 95% confidence level.

## 3 Results and discussion

Fluxes of GEM and GEM concentrations are shown in Figures 8a and 8b. Principally, we found GEM emission (positive fluxes) and a net mean emission of 8.9 ng m$^{-2}$ min$^{-1}$ over the 20 days. The largest measured deposition (negative flux) was 8.0 ng m$^{-2}$ min$^{-1}$, whereas the largest emission was 190.0 ng m$^{-2}$ min$^{-1}$. As expected, the large emission events were connected to increased wind speed and resultant increase in turbulent transport (Figure 9).

A rapid increase in GEM flux was found on April 30. Simultaneously, the pressure dropped rapidly from 1032 hPa to 1013 hPa and increased again to about 1025 hPa. During this abrupt pressure drop, latent and sensible heat fluxes decreased rapidly (Figure 7) and the temperature increased from about -18°C up to -4°C before decreasing to -13°C. Wind speed reached its maximum-recorded speed for the duration of the campaign during this event. At the same time, stability changed from unstable to stable conditions. The observations above indicate that this sudden increase in GEM flux is most likely explained by a front passing with a sudden change in meteorological conditions and changes in wind flow. We will consider this special case as an outlier. The meteorological parameters are shown in Figure 6.

At low temperature ($< -20\ \degree$C) only fluxes of GEM close to zero were present, see Figure 9b. Low temperatures are required for the occurrence of AMDE ($< -4\ \degree$C) (Lindberg et al., 2002;Skov et al., 2004), at which point GEM is oxidized to GOM. This indicates that GEM is so easily oxidized to GOM at lower temperatures ($< -20\ \degree$C) that GEM falls below the detection limit. This is consistent with the findings of Cole and Steffen (2010), Berg et al. (2003) and Cobbett et al. (2007), which found the

lowest concentrations of GEM ($<0.5$ ng/m3) at low temperatures ($<-15$) in spring after polar sunrise. Ozone and GEM depletions are correlated during AMDEs possibly due to reactions mainly with Br, and low temperatures favor the reaction between Br and GEM (Goodsite et al., 2004, 2012;Skov et al., 2004;Schroeder et al., 1998).

The measurements were started when depletion was already present and, as seen in Figure 8a and Figure 8b, depletion (low GEM concentration during April 23-25 (AMDE 1) and May 2-5 (AMDE 2)) was followed by GEM emission as observed by

Brooks et al. (2006), supporting that GEM is reemitted after AMDEs. The results correspond to the general understanding that GEM is initially removed rapidly from the atmosphere. This removal is most likely due to photolytic oxidation to oxidized mercury, which, contrary to GEM, has a very low surface resistance (Skov et al., 2006) and thus deposits relatively quickly. It is generally accepted that GEM production in snow is the result of a photochemical reduction of oxidized mercury to produce GEM. Thus, at first we hypothesized that oxidized mercury is reduced photolytically to GEM in the surface snow followed by

reemission. However, Ferrari et al. (2005) found that production of GEM is linked to the snow temperature and according to Steffen et al. (2015) and Steffen et al. (2002), the photochemical reduction of oxidized mercury in snow – and thus the reemission of GEM – is temperature dependent. Faïn et al. (2013) concluded that temperature and solar radiation were the main environmental parameters controlling GEM production in snow and found increased GEM production in snow even at snow temperatures below -5°C. Mann et al. (2015) found increased GEM flux from snow when the solar radiation and snow

temperature increased, even at low air (-20°C) and low snow (-15°C) temperature. Furthermore, they found in laboratory studies that temperature influenced Hg photoreduction kinetics when the snow is approaching their melting point ($>-2$°C), suggesting that temperature influences Hg photoreduction kinetics indirectly. Similarly, in the sub-arctic Dommergue et al. (2003) showed that melting snow emits more GEM than at lower snow temperatures. Therefore, an increase in atmospheric temperature and solar radiation increasing the snow temperature could lead to increased reemission of GEM causing the

concentration of GEM in the atmosphere to increase.

In the present study, the largest emissions were found during events with the highest temperatures (temperatures $> -15$°C), as seen in Figures 9b. The same behavior is not found for $CO_2$ flux (Figure 9c), where fluxes measured from -20°C to -15°C have the same magnitude as fluxes measured from -15°C to -10°C. The mean fluxes of GEM and $CO_2$ for the temperature intervals 5°C to -10°C, -10°C to -15°C and $>-20$°C also show an increase in the emission of GEM at increasing temperature (See Table

2), but a less clear relation between $CO_2$ flux and temperature. Both GEM and $CO_2$ fluxes correlate with the wind speed (Figure 10) and stability, thus we argue that the temperature could be a possible driver for the GEM emissions presented here. Oxidized mercury species are water-soluble, hence it is assumed that reduction of deposited Hg takes place in the aqueous phase (Steffen et al., 2015), which is followed by emission of the more volatile GEM. It is possible that the temperature relation observed in present study is due to an increased water content in the snowpack. Heating of the surface (i.e. downward sensible heat flux)

and upward latent heat flux (evaporation or sublimation) occurred on April 27 during the first larger GEM emission event (Event1), supporting the temperature- and water- dependency hypothesis. However, we found no strong relation between GEM flux and latent heat flux in general (See figure 11a), but we observed that high emission of GEM was in general associated with downward sensible heat fluxes (Fig 11b). A clear diurnal pattern for the radiation intensity was found, with the maximum at noon and the minimum at midnight, but these diurnal variations seem not to correlate with the GEM flux or concentration directly, see Figure 9a. Nevertheless, it is likely the snow is heated by the relatively strong solar radiation ($> 400$ wm$^{-2}$) during the day and by the air, when this is warmer than the snow. Unfortunately, we did not measure temperature or humidity in the snow, to support the suggested relation between emission, snow melting and air temperature in our study.

The increased concentrations of GEM may not only be caused by increased emission but part of the concentration increase could also be due to long-range transportation of GEM. Trajectory calculations of air mass transport on April 27 show downward mixing from higher elevations (Figure 3a), which could introduce air masses with higher GEM concentrations to our measurement site. However, at the same time we found upward fluxes of GEM, and in order to obtain an upward surface flux, the concentration in the snow must be higher than in the atmosphere.

The GEM emission on April 28 (event 2) was followed by an increase in GEM concentration on April 29. This occurred as the stability rapidly changed from stable ($z/L > 0$) to unstable ($z/L < 0$) conditions. The GEM concentration was relatively constant around 1 ng m$^{-3}$ on April 28 but increased threefold as the stratification changed from stable condition to unstable on April 29. According to trajectory calculations, this sudden increase was not caused by mixing from aloft (Figure 3b). We speculate that strongly stable conditions can result in GEM buildup directly above the surface, similar to $CO_2$ storage over forested sites (Yang et al., 2007). Surface emission of GEM into a relatively shallow layer of air will result in its higher concentration close to the ground. This buildup concentration would not be detected until the layer at the surface is mixed to a higher elevation when the stratification becomes unstable. On May 7, (Event 3) a change from stable to unstable conditions occurred simultaneously with an increase in concentration, which also partly could be explained by inversion of the surface layer as described above. The concentration increase was rapid, although not as large as the previous event (event 2), but the GEM emission in the days before event 3 were low and the stable conditions only lasted for a few hours (5-6 hours). Thus, we argue that the low GEM emissions lead to only a minor accumulation of GEM in the shallow surface layer before the surface layer was inverted. This concentration increase cannot be explained by a mixing from aloft as the trajectory calculations show a constant air mass transport pattern from May 3 to May 6 (Figures 3c and 3d), which should preclude such an event. There are other cases of stability change during our measurement period, but often the wind speed is higher, thus a shallow surface layer may perhaps not be formed. If a "build up" or "storage" effect exists, the flux measurements are also affected, and evaluation of flux data becomes even more complicated, thus, a more detailed study of the structure and dynamic of the Arctic atmospheric surface layers is needed.

This "shallow stable layer - inversion mechanism" is just a hypothesis, however, if this is a general pattern for very stable conditions, this can be an important effect, which needs to be considered in future measurements of Hg concentrations in the high Arctic. According to Osterwalder et al. (2016), GEM REA fluxes were significantly different under stable, unstable and

neutral conditions over a snow-covered surface. In the present study, GEM was primarily emitted under neutral and slightly stable conditions, and fluxes close to zero were observed under unstable and neutral conditions. On the other hand, Osterwalder et al. (2016) observed emission during unstable conditions, a small deposition during stable conditions and deposition during neutral conditions. The differences in emission during certain stabilities can be explained by a non-Arctic location and a very
different dynamic of GEM.

We observe some (anti)correlation between $CO_2$ and GEM from Figure 10. The correlation can be a result of the common correlation to wind speed, however, we speculate if chemical reactions or bacterial activity in the snow also could be part of the explanation of a correlation between the two fluxes; further research regarding this is needed.

In the following paragraphs, we compare our results to results found in other studies. We do not compare to studies using
chambers since this is a very different approach. Chamber measurements are enclosure methods, and therefore run the risk of potentially changing temperature, humidity, radiation, etc. (Fowler et al., 2001), furthermore chambers "capture" the exchange with the surface over a very small limited area. Micrometeorological methods, such as REA and AGM, are non-invasive and are thus more appropriate for comparing the results of the present study with other non-invasive methods.

Our findings do not agree with Cobbett et al. (2007) and Manca et al. (2013), as we found a few negative fluxes of GEM and
a large net emission of GEM during the campaign despite the potential for long-range transported GEM between April 25 and April 28. However, Brooks et al. (2006) report a small reemission of GEM with a net gain of mercury in the snow over a two-week period during March-April 2003 at Barrow, Alaska. A net emission of GEM was found in the present study, as well as in those conducted by Brooks et al. (2006) and Steen et al. (2009), but the net GEM flux evident in the present study is much higher than others have observed. A study by Ferrari et al. (2004) was performed at the same location as the present study, but
the range of the fluxes found was more than three orders of magnitude lower than presented here, maybe due to higher wind speeds and concentration levels during the present study. Despite the difference in magnitudes of fluxes, GEM depletion was observed in all three studies. Brooks et al. (2006) estimated the GOM flux from surface resistance models based on results in Skov et al. (2006), while gradient measurements were used to estimate the GEM flux, thus the difference in the estimated fluxes can also be explained by differences in the methods used. Measurements by Cobbett et al. (2007) from April to June in
Alert, Canada showed zero net flux. The most significant fluxes observed during polar day were found in early June when the soil was visible, which was never the case during the present study's campaign. Manca et al. (2013) found a net deposition at Ny Ålesund, Svalbard from April to May with significant depositions and emissions, which can be explained by the location, since Ny Ålesund is located at open seawater and thus it is not expected that any local AMDEs would take place because AMDEs are related to sea ice and snow surfaces. During the present study, the air masses recorded derived mainly from sea
ice during the depletion events (Figures 4a and 4b) in spite of the local SW winds.

As mentioned earlier, we speculate that strongly stable conditions can result in GEM accumulation directly above the surface. Brooks et al. (2006), Cobbett et al. (2007) and Manca et al. (2013) all used the flux gradient method to determine GEM flux, and the different results obtained could be due to the flux measurement techniques used. Using the gradient method, flux is

estimated from concentration measurements at different heights. Strong stratification with GEM buildup near the surface will likely result in a non-constant flux layer, violating a basic assumption for the flux gradient method.

The study sites in the present study and in the studies by Brooks et al. (2006), Cobbett et al. (2007) and Manca et al. (2013) differ significantly in terms of orography and meteorology, which have an effect on the fluxes. Theoretical studies by (Goodsite et al., 2012, 2004) show that GEM removal is driven by chemical reaction with Br and increases with decreasing temperature. The differences in locations, orography and meteorology between research sites affect the concentrations of GEM, because parameters such as temperature, Br concentration and origin of air masses are different for the sites. The wind direction in the present study was primarily from SW, caused by katabatic winds from the local Flade Isblink ice sheet; however this is merely the source of the local wind and most air masses in the study area overall are derived from sea-ice covered surfaces according to the trajectory calculations (see Figures 3 and 4). As mentioned, atmospheric stability influences the observed GEM fluxes (Osterwalder et al., 2016) and different stability conditions between sites could explain the differences in fluxes found by Cobbett et al. (2007) and Manca et al. (2013). Overall, our results suggest that variations in GEM concentrations and fluxes are much more variable than previously assumed.

## 4 Conclusion

Mercury is primarily transported in the atmosphere in the form of GEM and it is ubiquitous in the atmosphere. Fluxes of GEM have been measured at Villum Research Station, Station Nord, in the high Arctic of north Greenland over snow-covered surfaces from April 23 to May 12, 2016 with a REA system utilizing dual inlets and dual detectors.

This work showed an average GEM emission of 8.9 ng $m^{-2}$ $min^{-1}$ during the 20-day research campaign, during which several AMDEs were observed. A maximum deposition of 8.0 ng $m^{-2}$ $min^{-1}$ and a maximum emission of 190 ng $m^{-2}$ $min^{-1}$ were recorded. The results of this study support to some extent the general understanding of the AMDE mechanisms where GEM oxidation is followed by deposition of GOM, which is partly reduced to GEM and reemitted into the atmosphere. Furthermore, the data show some relation between increase in upward GEM fluxes and increasing temperature and heating of the snow surface. However, the scatter on the flux data is large and the snow temperature is not measured in present study, thus further detailed studies to investigate this relation is needed.

The observed fluxes and concentrations are related to meteorological conditions and comparing concentrations and fluxes found at other high-latitude sites reveals wide variation between sites. However, these comparisons imply that GEM fluxes and concentrations can be rather heterogeneously dispersed in the Arctic atmosphere due to the complex meteorological flows and stratification.

Further studies on this heterogeneity, including potential inversion at the surface and mixing from aloft, are needed, as are studies of fluxes of both GEM and GOM adjacent with measurements of the energy budget and controlling parameters extant in snow pack.

## Competing interests

The authors declare that they have no conflict of interest.

## Acknowledgements

The Department of Environmental Science, Aarhus University is acknowledged for providing logistics at Villum Research Station in North Greenland. "The Danish Environmental Protection Agency" financially supported this work with means from the MIKA/DANCEA funds for Environmental Support to the Arctic Region and by the Arctic Research Centre (ARC), Aarhus University. The military personnel at Station Nord are acknowledged for their help during the measuring campaign. The authors gratefully acknowledge the NOAA Air Resources Laboratory (ARL) for the provision of the HYSPLIT transport and dispersion model and/or READY website (http://www.ready.noaa.gov) used in this publication. This work is a contribution to the Arctic Science Partnership (ASP). We thank the anonymous reviewers for their valuable comments and suggestions to improve the quality of the paper.

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

**Figures and tables**

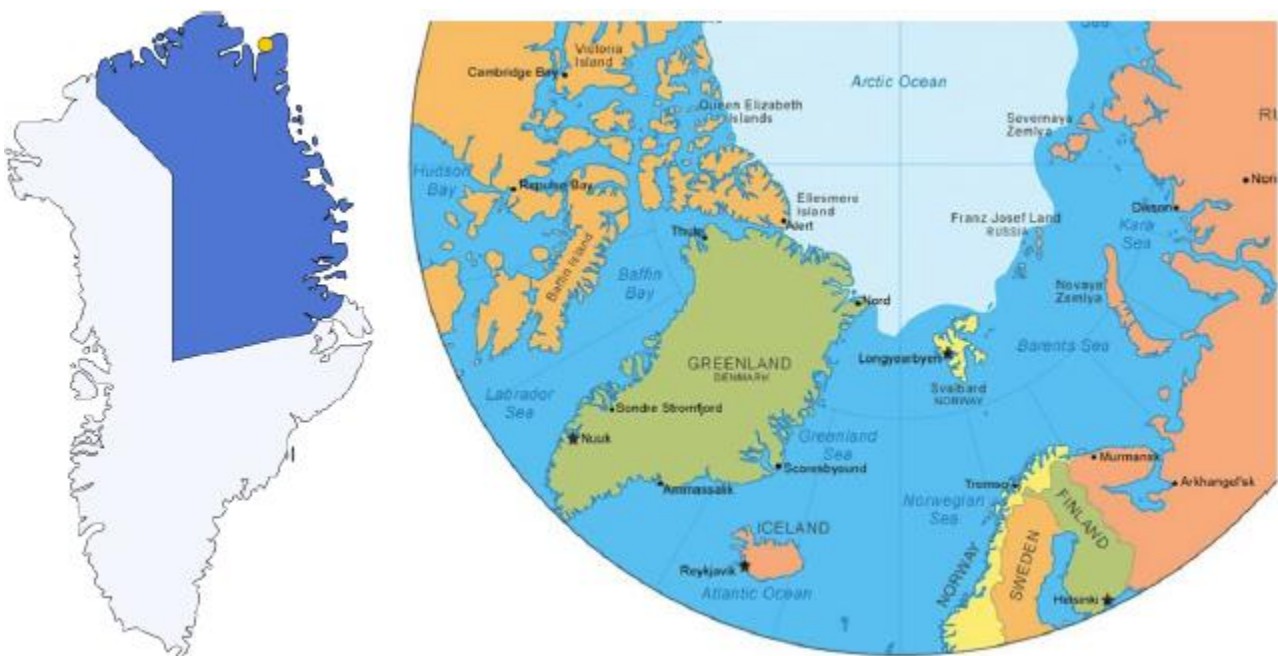

**Figure 1: Left: Greenland with indication of the largest Natural Reserve in the world (blue) and the position of Station Nord (yellow dot). Right: The northern hemisphere where Station Nord (Nord) also can be seen.**

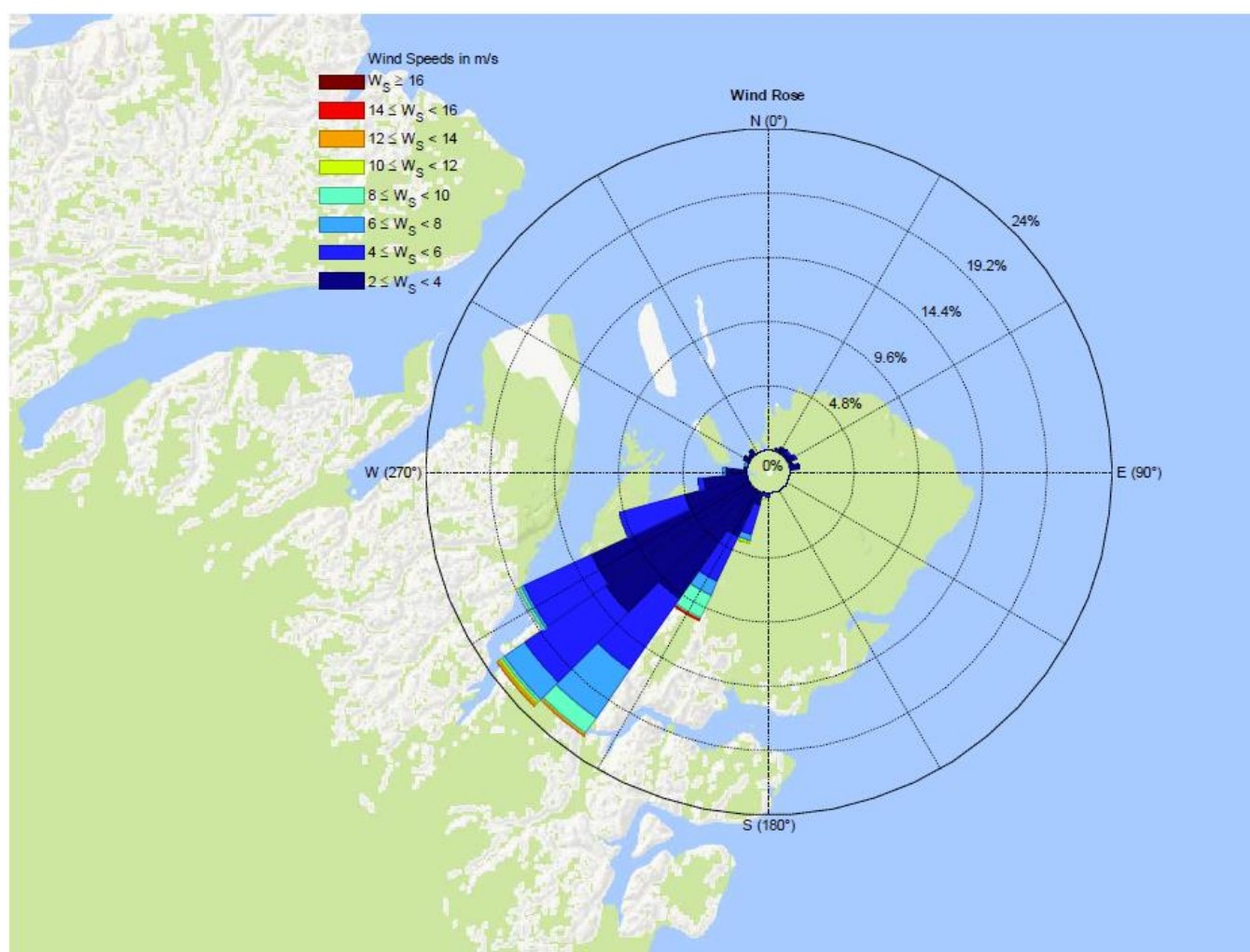

**Figure 2: Wind rose centered at Villum Research Station, Station Nord. Length of the bars indicate frequency of the direction and color indicate the wind speed. Units in m s⁻¹.**

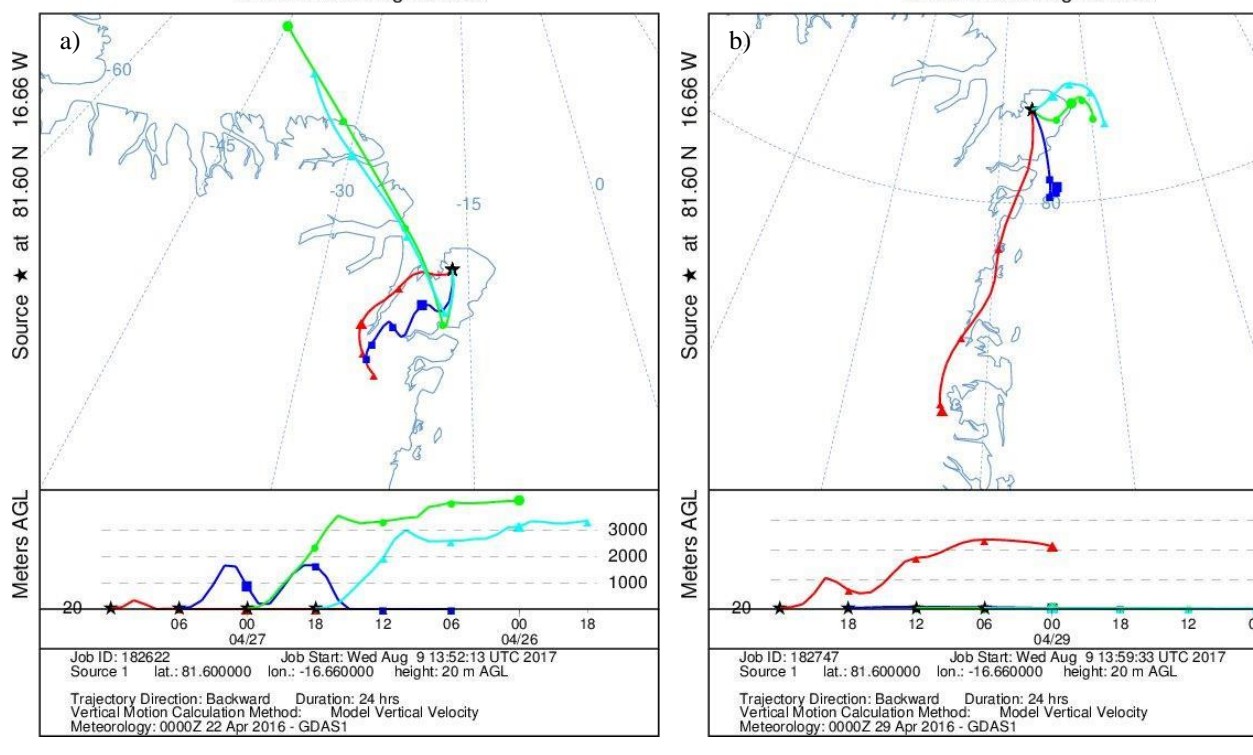

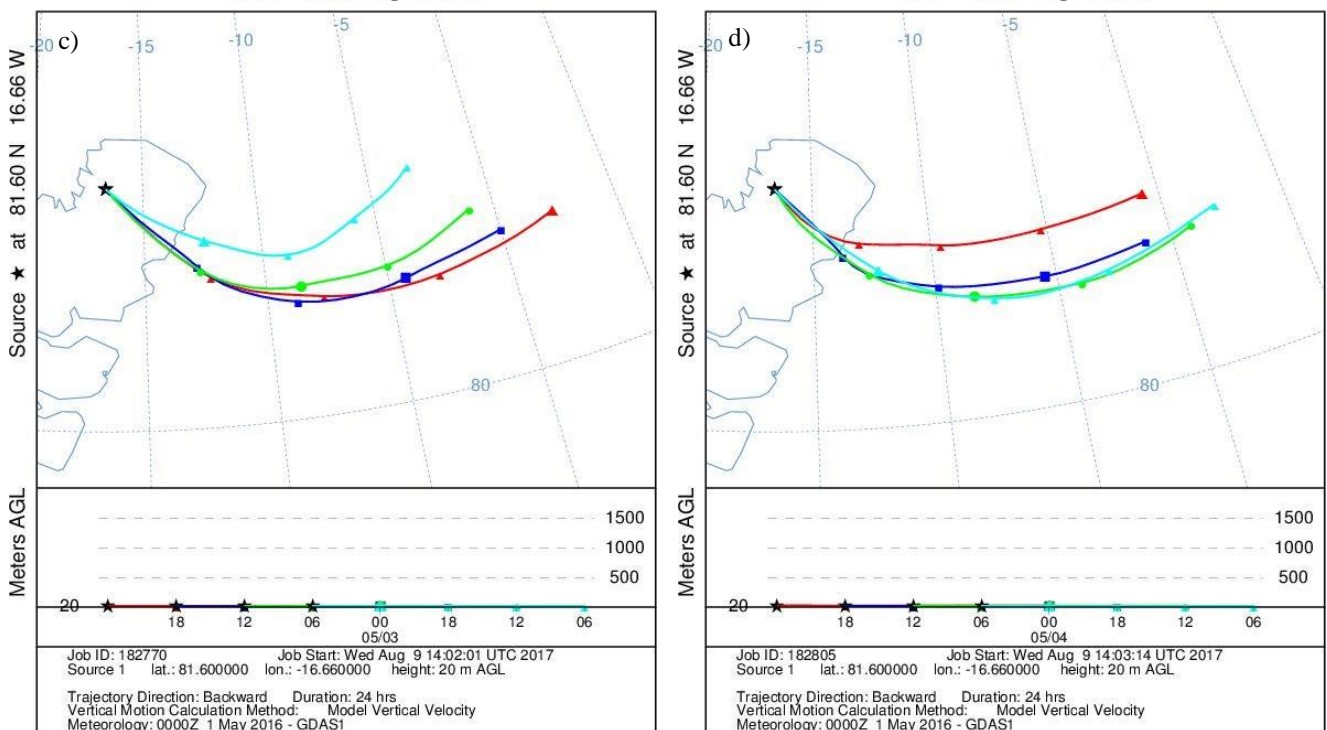

**Figure 3: Backward trajectories for four different days showing the origin of the air masses. All trajectories are 24-hour calculations and each figure show a new trajectory for every 6 hour backward. a) Starting April 27, 12:00. b) Starting April 30, 00:00. c) Starting May 4, 00:00. d) Starting May 5, 00:00.**

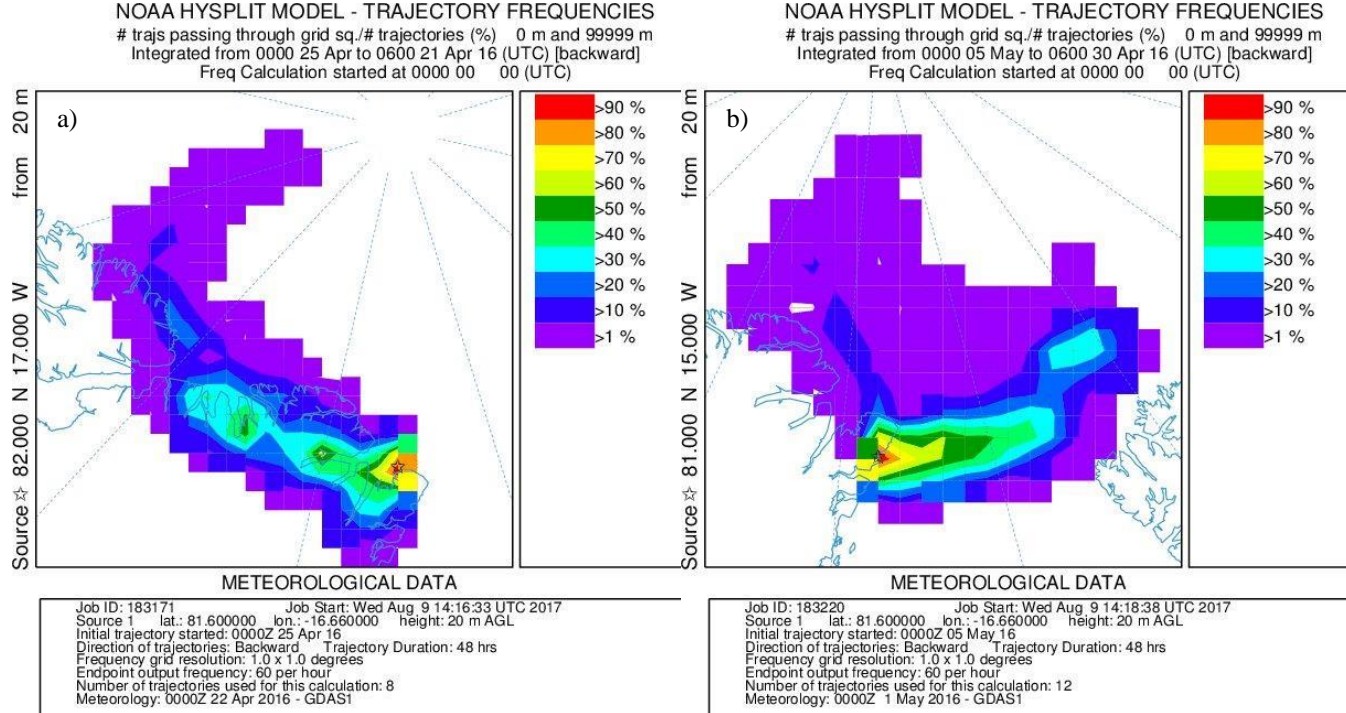

**Figure 4: Trajectory frequencies showing the number of trajectories passing through a grid. The resolution is 1 degree. a) April 21 to April 25. b) April 30 to May 5.**

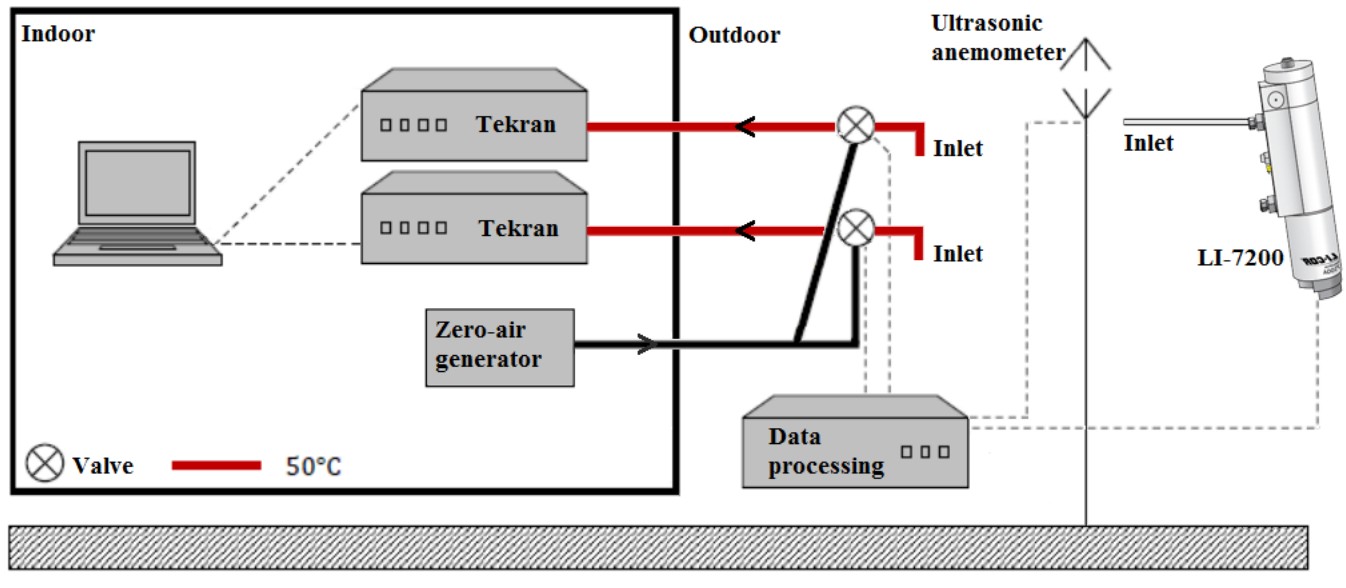

5    **Figure 5: A Schematic representation of the GEM REA system.**

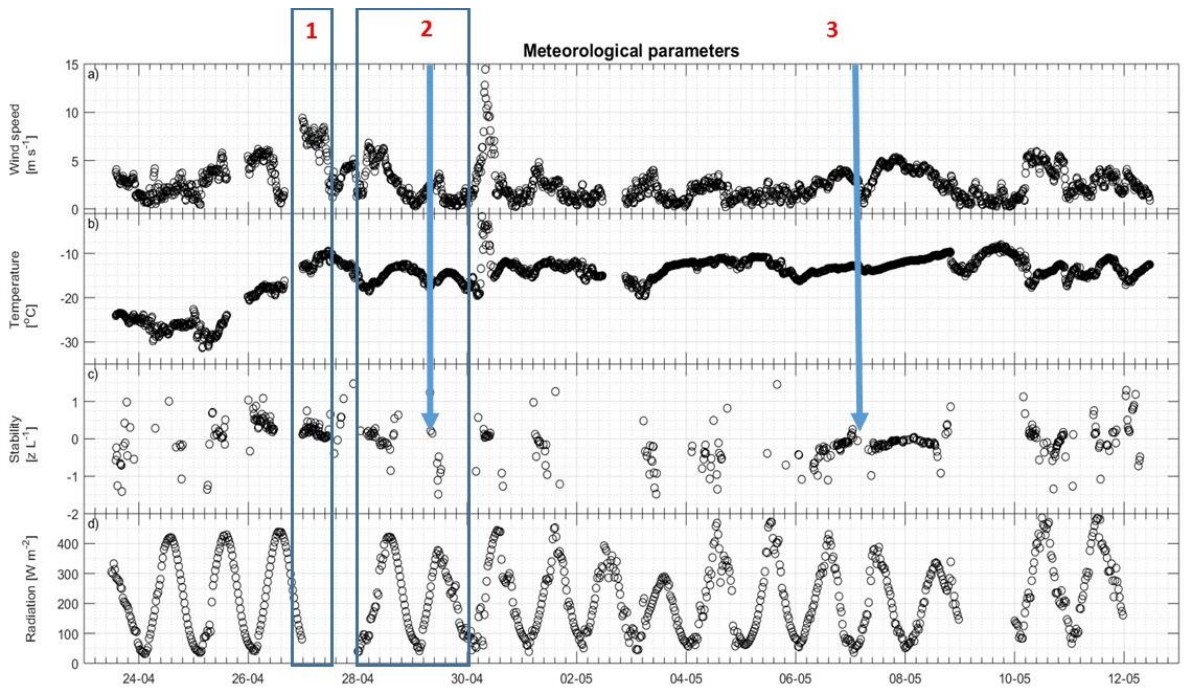

**Figure 6: Meteorological parameters over time with three events highlighted. The arrows show the specific case of stability change referred to in the text. a) Wind speed in m s⁻¹, b) Temperature in °C, c) Stability as z/L and Radiation as W m⁻².**

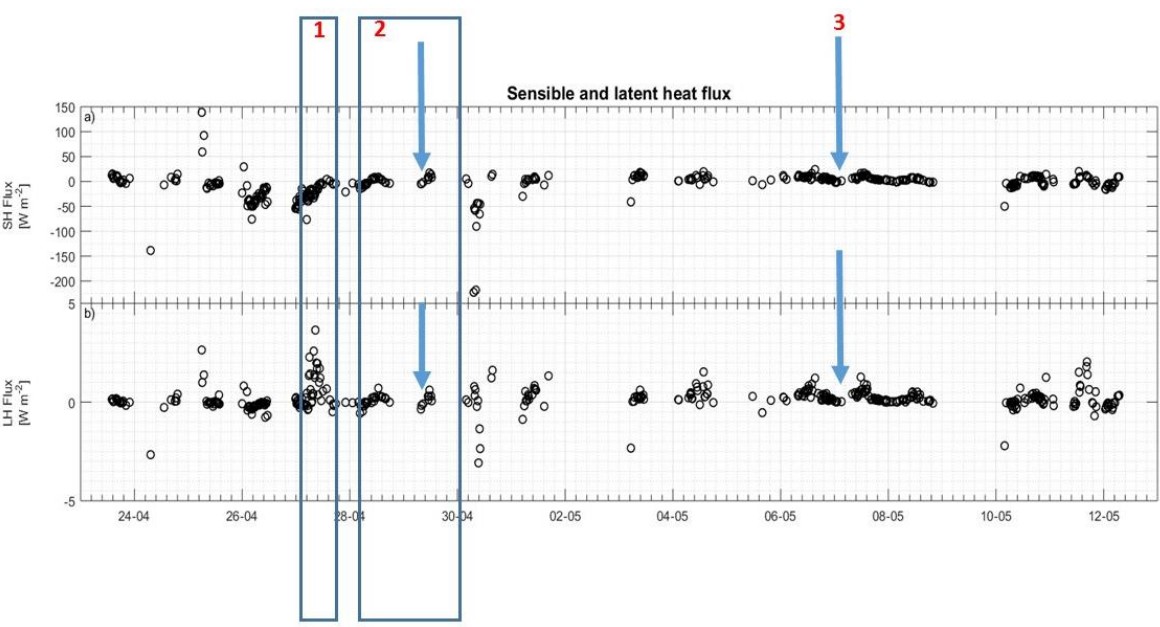

5    **Figure 7: Sensible and latent heat flux in panel a) and b), respectively. Both are measured in W m⁻² and three events are highlighted. The arrows show the specific case of stability change referred to in the text.**

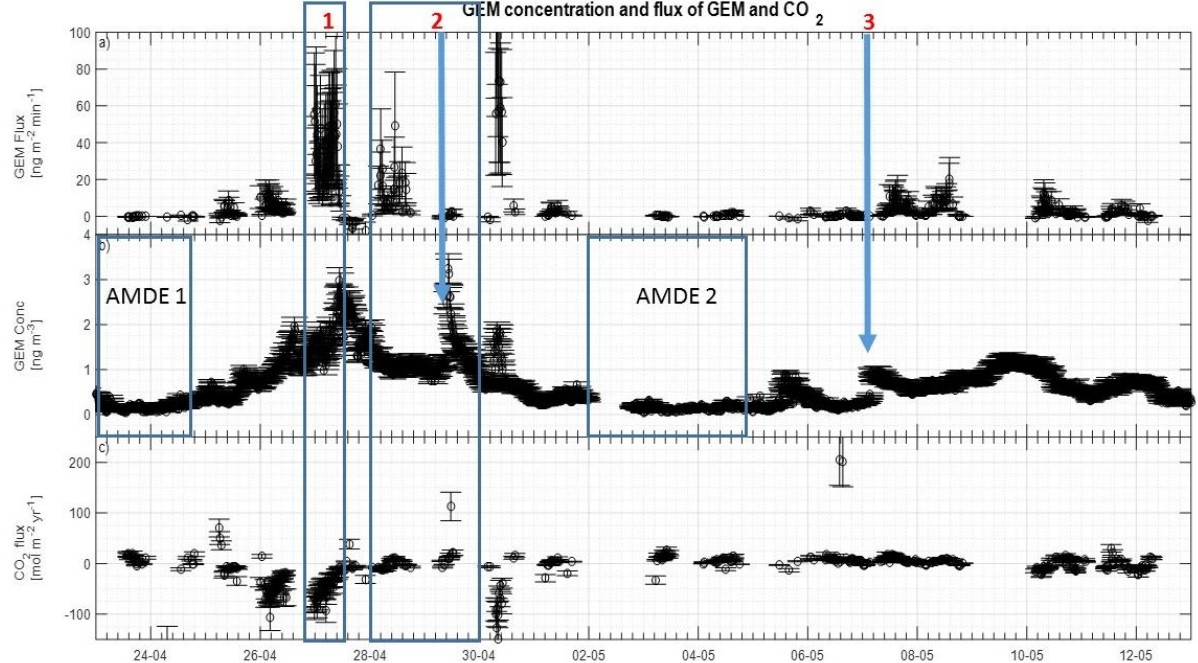

**Figure 8: GEM concentration and fluxes of $CO_2$ and GEM over time. a) GEM flux in ng m$^{-2}$ min$^{-1}$, b) GEM concentration in ng m$^{-3}$, and c) $CO_2$ flux in mol m$^{-2}$ yr$^{-1}$. Three events and two AMDEs are highlighted. The arrows show the specific case of stability change referred to in the text.**

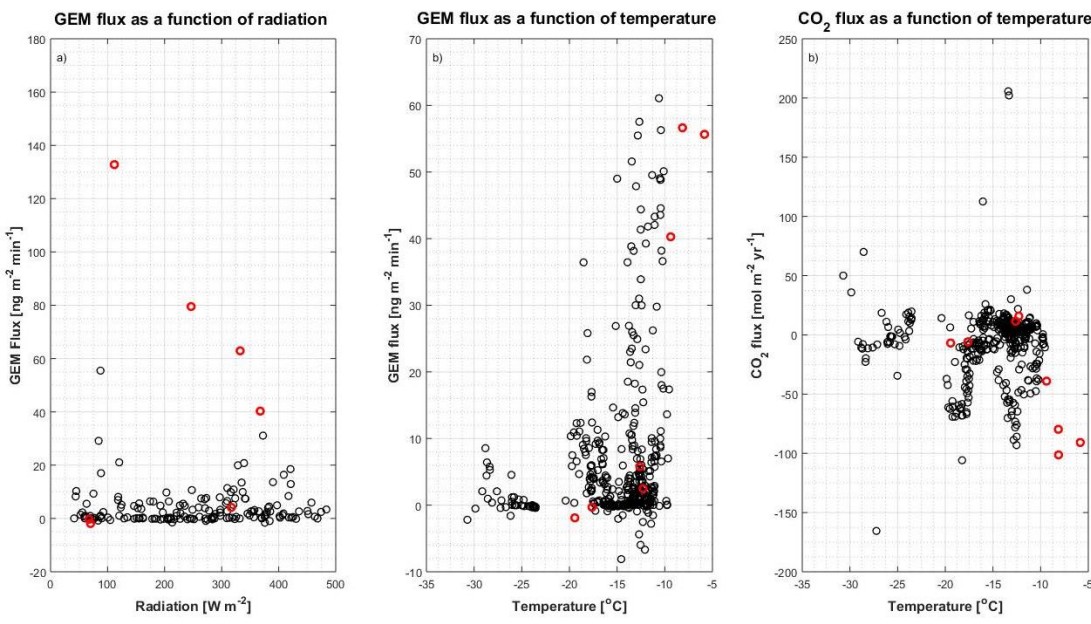

**Figure 9: Relation between GEM and $CO_2$ flux and temperature. a) GEM flux in ng m$^{-2}$ min$^{-1}$ as a function of solar radiation, b) GEM flux in ng m$^{-2}$ min$^{-1}$ as a function of temperature, and c) $CO_2$ flux in mol m$^{-2}$ yr$^{-1}$ as a function of temperature. The red circles are data from the period on April 30, which are considered as outliers (see the text for further explanation).**

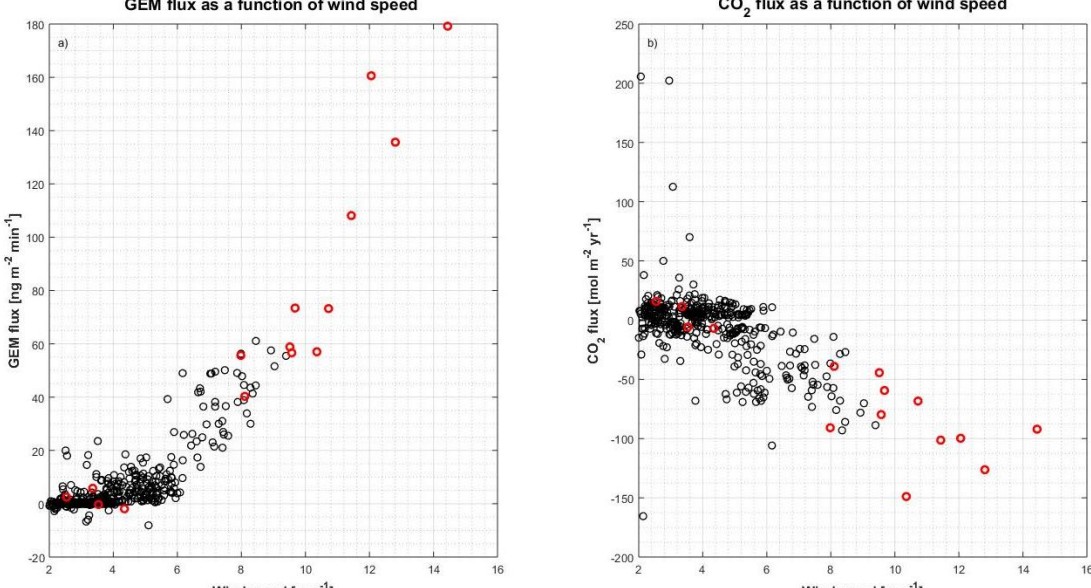

**Figure 10: Relation between GEM and $CO_2$ flux and wind speed. a) GEM flux in ng m$^{-2}$ min$^{-1}$ as a function of wind speed, and b) $CO_2$ flux in mol m$^{-2}$ yr$^{-1}$ as a function of wind speed. The red circles are data from the period on April 30, which are considered as outliers (see the text for further explanation).**

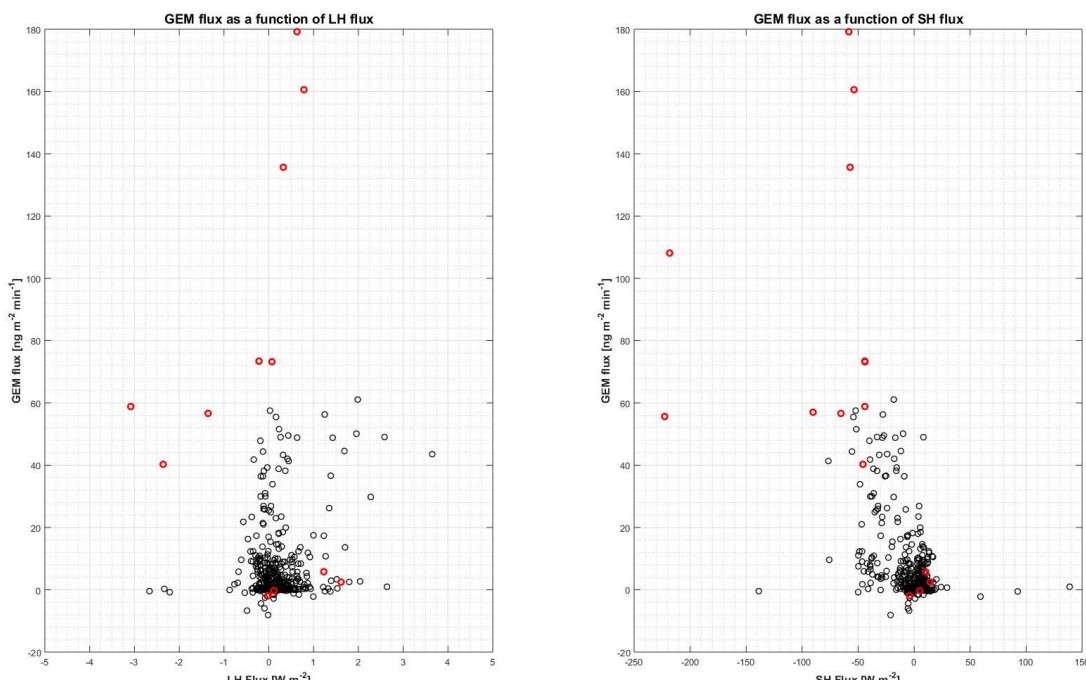

**Figure 11: Relation between GEM flux and heat flux. a) GEM flux in ng m$^{-2}$ min$^{-1}$ as a function of Latent heat flux in W m$^{-2}$, and b) GEM flux in ng m$^{-2}$ min$^{-1}$ as a function of sensible heat flux in W m$^{-2}$. The red circles are data from the period on April 30, which are considered as outliers (see the text for further explanation).**

**Table 1: Summary table over reported GEM fluxes in the Arctic. The units are changed from those within the references for better comparison.**

| Flux | Site | Method | Reference |
|---|---|---|---|
| Mean: 8.9 ng $m^{-2}$ $min^{-1}$<br><br>Range: -8.1-179.2 ng $m^{-2}$ $min^{-1}$ | Villum Research Station, Station Nord, Greenland | Relaxed eddy accumulation | Present study |
| Mean: 0.050 ng $m^{-2}$ $min^{-1}$<br>(in reference: 1.0 $\mu$g $m^{-2}$ 14 $days^{-1}$) | Barrow, Alaska | Flux gradient method | Brooks et al. (2006) |
| Mean: -0.60 ng $m^{-2}$ $min^{-1}$ | Alert, Canada | Flux gradient method | Cobbett et al. (2007) |
| Mean: -0.004 ng $m^{-2}$ $min^{-1}$ | Ny Ålesund, Svalbard | Flux gradient method | Manca et al. (2013) |
| Median: 0.12 ng $m^{-2}$ $min^{-1}$ | Ny Ålesund, Svalbard | Flux gradient method | Steen et al. (2009) |
| Range: 0.001-0.007 ng $m^{-2}$ $min^{-1}$ | Station Nord, Greenland | Flux gradient method | Ferrari et al. (2004) |
| Range: 0-0.8 ng $m^{-2}$ $min^{-1}$ | Ny Ålesund, Svaldbard | Flux chamber | Ferrari et al. (2005) |
| Max: 0.58 ng $m^{-2}$ $min^{-1}$ | Ny Ålesund, Svalbard | Flux chamber | Ferrari et al. (2008) |
| Mean: 0.13 ng $m^{-2}$ $min^{-1}$ | Ny Ålesund, Svalbard | Flux chamber | Sommar et al. (2007) |

5 **Table 2: Mean and standard deviation of GEM and $CO_2$ fluxes in different temperature ranges calculated from the data sampled at Station Nord from April 23 to May 12 in 2016**

| Temp. range (°C) | Mean GEM Flux (ng $m^{-2}$ $min^{-1}$) | Mean $CO_2$ flux (mol $m^{-2}$ $yr^{-1}$) |
|---|---|---|
| < -20 | 0.93 ± 2.26 | -0.31 ± 32.39 |
| -20 to -15 | 5.13 ± 6.28 | -16.54 ± 30.15 |
| -15 to -10 | 8.40 ± 14.20 | -2.57 ± 28.53 |
| -10 to -5 | 24.96 ± 33.95 | -28.92 ± 38.88 |