# Peer review of "Fluxes of gaseous elemental mercury (GEM) in the High Arctic during atmospheric mercury depletion events (AMDEs)"

_Atmospheric Chemistry and Physics, 2017_

## Referee Comment (RC1) · Anonymous Referee #2 · 13 Sep 2017

**General comments**:

The authors present a first paper using the relaxed eddy accumulation (REA) method to measure flux over an Arctic snow covered surface, providing useful results for those looking to undertake similar Arctic mercury flux studies in the future. However, thorough editing of the text for grammar and clarity should be undertaken; in many areas, sentence fragments are present (see technical corrections for examples), and this should be corrected prior to publication, to increase the comprehensibility of the text. In addition, I would also caution that the authors appear to be extrapolating their conclusions from a relatively limited data set, and many of the conclusions drawn from their results do not appear to be adequately supported by the presented results. In their Fig. 7a, GEM fluxes appear to be predominantly near 0, which means that any derivation of relationships based on this data set must be approached with caution, and broad sweeping statements about the importance of various factors should be avoided (see specific comments below).

**Specific comments**:

1. *Use of the REA method*:

The manuscript presents a study using a technique (REA) which had not previously been used for this type of Arctic GEM flux work; however, the authors fail to explain why they have chosen this technique. In a "first use" study such as this it is important to explicitly outline the benefits of this technique over more traditionally used or other available methods, as well as the potential short-comings. In its present state, it is unclear why this work using the REA method is a benefit to the Arctic Hg flux literature, or why one may choose to do a REA based Hg flux study rather than use a chamber or AGM type method.

2. *Exclusion of data (primarily $CO_2$ flux to determine b section, and one Results reference)*:

It is not immediately clear how the limits were chosen for various parameters to allow for inclusion or exclusion of data (in "$CO_2$ flux to determine b" section). In total, the authors state that they excluded 74% of the collected data, which seems rather extreme, especially lacking adequate justification for the exclusion criteria. The authors state that they have estimated an uncertainty for b ($<<10\%$), and state that the uncertainty is then assumed to be insignificant, but provide no methodology for that estimation, or justification for the assumption of insignificance. In this section, the authors also include a "z/L" criteria as a means by which data were included/excluded from the presented results, but this "z/L" is neither defined, nor described. Various parameters are given, however the choice to discard data should be more thoroughly explained and justified, especially when it means that very little of the originally collected data is included and used to derive the final relationships presented in this work. In the Results and Discussion (pg. 7, line 18 – 19), the authors propose a means by which some data may be falsely

interpreted as outliers, but it is unclear whether this was a problem in the presented data set, and, if so, how was this dealt with?

3. *Uncertainty in presented data (pg. 6, lines 12 – 14)*:

The authors provide an estimate of the uncertainty "of the two concentration determinations", but it is unclear what precisely they mean by "the two concentration determinations"; is this the uncertainty of Hg flux based on the instrumental detection limit of 0.1 ng/m$^3$? If so, this needs to be clarified (and units should be included with the appropriate numbers). In addition, it is stated that the uncertainty of the flux (which is given as 0.14) becomes 28% at the 95% confidence interval (unclear how this was derived), and then in the following sentence the authors give a GEM uncertainty of 10% above 0.5 ng/m$^3$, but this is attributed to another study, so it is unclear how this fits in with any of their data, or uncertainty in their data, and precisely what the uncertainty on any of the provided results would be. In addition, no factors besides flux and b are given any consideration with respect to uncertainty, which makes it impossible to know how robust any of the presented results/relationships are.

4. *Results and Discussion*:

I believe that this section of the manuscript should be thoroughly reviewed by the authors, and revised prior to publication, as many conclusions/relationships appear to have been drawn from GEM flux results that, as presented, typically seem to be approximately zero (Fig. 7a). In instances where the authors are calling on certain events (eg/increases in flux) these events should be explicitly stated, for clarity.

5. *The importance of heat/temperature changes*:

On pg. 2 (lines 30 – 31) the authors propose that a correlation between solar radiation induced heat flux/temp change and GEM production/flux exists, due to the observed relationship between solar radiation and GEM concentration increases in surface snow, and then again in the results/discussion (pg. 8, lines 9 – 12) they propose a temperature dependence in their observed results, and in the conclusions (pg. 9, lines 29 – 31) state that their data support the hypothesis that heating of the surface is influencing GEM formation/emission; however, this ignores the generally accepted idea that GEM production in snow (and many other media) is the result of photochemical reduction of $Hg^{2+}$ to produce $Hg^0$. It is possible that this process ($Hg^{2+}$ reduction or subsequent $Hg^0$ emission) from snow may be influenced by heat, but the primary reason for an increase in $Hg^0$ production with increased solar radiation intensity will most likely be due to increases in the extent of $Hg^{2+}$ photoreduction, not as a result of heat induced processes in the snowpack.

In addition, in the results/discussion section, the authors state "After deposition, we speculate that GOM is reduced photolytically to GEM." (pg. 6, line 23 – 24), but then later state "GOM is reduced at the snow surface when temperature increases (Ferrari et al., 2008)." (pg. 6, line 26).

GOM in the snow will most likely be reduced by some manner of photochemical reduction reaction (whether this be photolytic, or as a result of some other photochemically driven process), and while temperature may influence the extent of this reaction, or movement of GEM from the snowpack, radiation (sunlight) is required for the reaction to proceed. As it is used in the aforementioned statement, the Ferrari et al. (2008) reference is somewhat misleading, and this should be revised. In addition, the authors are using air temperature rather than the snowpack surface temperature to derive relationships with flux, and these will differ. Since the surface snow temperature may be significantly different than the ambient air temperature, it is difficult to make compelling conclusions about temperature dependence on snowpack Hg flux without these snowpack temperatures.

In addition, the authors state that "The highest temperatures were found during events with the largest emissions…" (pg. 7, line 32 – 33), but these events have not been explicitly pointed out to the reader using the dates provided in the figures. From the data, I can see three easily observable GEM flux increase events (April 27, 28, 30); however, in one of these three easily observable emission events (April 28) increasing GEM fluxes occur before temperatures begin to increase, and, at least at the beginning of the GEM flux increase event, temperature is actually decreasing, and this is counter to what the authors have stated in their text. As a result, the text should be revised to explain this divergence from what they are stating to be typical behaviour (if more than the three observable events are being invoked).

The authors use an instance of an upward latent heat flux (presumably indicated as a positive number in Fig. 7e?) on April 27 which coincided with a GEM emission event as evidence to support their GEM flux temperature/water dependence hypothesis; however, this increase in LH flux appears to be quite small, and while it does coincide with an increase in GEM flux, other increases in LH flux which appear to be comparable in magnitude (eg/April 25 – 26) did not result in increased GEM fluxes. Further, the second observed increase in GEM flux (April 28) appears to have occurred with no significant change in LH flux, and the largest observed increase in GEM flux (April 30) occurred with a LH flux decrease (negative value) that was much more significant than any of the LH flux increases observed in the data set. At best, it would appear that the data presented by this work appears to neither support nor dispute the hypothesis of temperature dependence of Hg flux based on LH flux information, but it supplies no compelling evidence to support it. If the latent heat flux argument is to be included, other incidents which were counter to the authors' hypothesis must also be discussed in the text, and adequate reasoning provided as to why these do not give evidence to disprove the proposed hypothesis.

6. *The effects of wind speed on GEM flux*:

The authors state that all large emission events occurred when wind speeds increased (pg. 7 line 10), but it is unclear how wind speed effects and temperature effects are distinguished, or whether the proposed temperature effect on flux (see above) is simply the result of greater wind speeds appearing to coincide with increases in temperature (Fig. 6). With so many variables

changing (potentially independently) at the same time, it is not possible to tease apart the relative importance of these on GEM flux by visual observation alone, which is how the proposed relationships appear to have been derived.

7. *Conclusions based on flux data trends*:

The authors attempt to determine relationships and draw conclusions regarding the effects of various factors on GEM flux; however, in looking at the data in Fig. 7a, it appears that in most instances, GEM flux is at or very near 0 ng/m$^3$; while this may be a function of the scale being used, there simply does not appear to be adequate data to support strong conclusions, with the data presented, especially with no inclusion of the uncertainty on these data. Further, if the authors plan to make conclusions regarding the dependence of flux on the various factors that they measured, it would be very helpful to have some manner of statistical test to back up these claims, as without them, the invoked trends are neither clear nor compelling. For example, the authors state that depletion events on April 23 – 25 and May 2 – 5 are followed by GEM emissions; however, while this may be supported by the April 27 increase in GEM flux that is visible in the results, there does not appear to be a significant increase in GEM flux following May 5, until the small (almost apparently negligible) increase in flux on May 7, when GEM concentrations are higher again. With the data as it has been presented, this does not appear to support GEM emission post-AMDE as the authors have proposed (pg. 6 line 21).

8. *Correlation between $CO_2$ and GEM*:

The authors state that there is a correlation between $CO_2$ and GEM, but offer no methods used to determine this. Was this decided based on mathematical/statistical analysis? Simple observation? Is this GEM concentration, or GEM flux? Based on a quick visual inspection of the results in Fig. 7a/b and 7c, there does not appear to be a compelling case for simple visual observation of such a trend. If this conclusion is to be included in the paper, there should be a more thorough investigation of the claim, or the methods used to draw this conclusion should be explicitly stated, at the very least.

9. *Comparison of study results with literature results (pg. 7 lines 19 – 24, pg. 8 line 15 – pg. 9 line 21)*:

Overall, the authors' discussion of their results as compared to other literature results is somewhat difficult to follow, and should be revised for clarity. This portion of the manuscript would benefit from the inclusion of more concrete results and explicit discussion (eg/ pg. 7 line 23 – 24: how can your results being in opposition to those from the Osterwalder study be explained by the difference in location? How is the GEM dynamic different in these studies? eg/pg. 8 line 21: what was your net emission, exactly? What were the values found by the other works that are referenced?), and from complete discussion of one topic before moving on to another to improve flow and comprehensibility (eg/ stability conditions are discussed in more than one place). Overall, as a discussion paper, comparison with other studies should be much

clearer, allowing the reader to easily place the present study with those already existing in the literature (does it agree with other studies using similar method or not, and why?), and at present this is not the case. It is further unclear, in some instances, how certain discussions relate to the present study. For example, when discussing stable conditions/GEM build-up (pg. 9, lines 3 – 10), the authors state that strong stratification with a build-up of GEM near the surface will result in violation of a basic assumption for the flux gradient method; was this phenomenon expected in your study, and if so, how did you deal with it? If this violation of a basic assumption for the flux gradient method was not observed in your study, why have you included it here? The discussion section (and comparison to other studies) might also be easier to follow if a better introduction to the chosen technique was included (see specific comment #1).

10. *Results figures/tables*:

For the results figures (Fig. 6 and 7), the markers/scales chosen make it almost impossible to see differences in the data with time, except where those differences are very large. Since the authors are attempting to use such differences in various measurements over time to derive information regarding factors influencing GEM flux, it is imperative that the reader be able to see the differences the authors appear to be speaking of, and at present, this is not true in most cases. In addition, where uncertainty is known (eg/flux uncertainty = 28%, as stated in the manuscript) error bars should be provided for the data in these figures, as it is unclear whether changes (eg/in flux over time) might be statistically significant, or not. Also, certain events (eg/ AMDEs as in pg. 9 line 26 – 27: "…during which several AMDEs were observed.") should be explicitly marked on your data, or the dates you are proposing they have occurred should be present in the text and/or figure captions.

In the summary table (Table 1, pg 21) the authors give a flux range of 8 – 190 ng m$^{-2}$ min$^{-1}$ for their data set; however, in looking at the results in Fig. 7a, it is apparent that there were some incidents of negative (depositional) flux (April 27), and there are many instances where the flux appears to be zero. As a result, it appears that the flux range given in the table is either incorrect, or some values were excluded, and if they were excluded, a reason should be given for this, as the table is not particularly informative without it.

**Technical corrections**:

Pg. 1 line 27 – pg. 2 line 2: These two sentences appear to be contradictory, and it is unclear what the authors are attempting to inform the reader of re: atmospheric lifetime of GEM with the given text.

Pg. 1 line 29: What do you mean by "…the relaxation time of mercury in the atmosphere…"? Please clarify/revise.

Pg. 2 line 4 – 5: Sentence fragments; consider adding fragment "These atmospheric mercury depletion events…" to the previous sentence.

Pg. 2 line 8 – 9: "Thus, this is a human health…" sentence fragment.

Pg. 2 line 13: Consider revising "…concentration decreases during…" to "…concentration decreases due to…"

Pg. 2 line 18 – 19: "Vertical gradient…" is repeat of the information presented in previous sentence, consider revising.

Pg. 2 line 22 – 24: "This is likely due to…" sentence fragment, consider adding to the previous sentence.

Pg. 2 line 25: affect, not affects.

Pg. 2 line 30 – 32: "Thus it is likely that a correlation…" this statement is somewhat misleading, as increases in solar radiation lead to increases in GEM production/emission as a result of an increase in $Hg^{2+}$ reduction, which may have nothing to do with heat!

Pg. 5 line 19: The detection limit you have given is the literature value for the instrumental detection limit, which may be significantly different than your method detection limit. Consider revising to be the method detection limit.

Pg. 5 line 25: "EC" is used without definition in the text (provided in the abstract, but should also be included on first use in the main body of the manuscript).

Pg. 5 line 26 – 28: "The flux of $CO_2$…" Sentence is confusing, please revise.

Pg. 6 line 2: Was b derived based on T (presumably temperature?)? If so, this method has not been described.

Pg. 6 line 7: What is z/L? This is not defined.

Pg. 6 line 26 – 27: "This leads to increased…" Sentence fragment, please revise.

Pg. 6 line 31 – pg.7 line 2: "We observed a clear diurnal pattern…" Did you measure incident solar radiation intensity? If so, this data should be included, and if not, how have you arrived at this conclusion/the timing of max and min sunlight?

Pg. 7 line 15 – 18: "This could have occurred…" Sentence fragment, please revise.

Pg. 7 line 23 – 24: "These differences can be explained…" Sentence fragment, please revise.

Pg. 8 line 3: "At low temperature (< -20 C) the flux of GEM was near zero." GEM flux was also near zero in many cases when the temperature was > -20 C; this should be discussed.

Pg. 8 line 24 – 25: "This could be due to higher wind speeds…" Sentence fragment, please revise.

Pg. 8 line 31 – 34: The two sentences about the Manca et al. (2013) study should be combined, as the second is mostly redundant.

Pg. 9 line 14 – 15: "Differences in locations between research sites…" Please state why this is important and/or how this is expected to influence your study with relation to others, as there is no context for this statement at present.

Pg. 9 line 15: "Important parameters are…" Sentence fragment, please revise.

Pg. 9 line 29 – 31: I don't believe you have provided a compelling case to support the given hypothesis, as the manuscript stands.

Pg. 14 Fig. 1: It's a little bit difficult to find the yellow dot in your figure, consider giving this dot a dark coloured outline to increase visibility where it overlies the white page.

Pg. 15 Fig. 2 caption: Should be "indicates", not "indicate".

Pg. 16 Fig. 3 caption: Should be "shows" not "show".

Pg. 19/20 figure labels: authors are not consistent with the way they are writing units in the text vs. the figures (eg/ $ng/m^3$ in text vs. $ng\ m^{-3}$ in figures).

Pg. 20 Fig. caption: "mol" not "mole".

Pg. 21 Table title: Should be "Summary table of…" rather than "Summary table over…"

---

## Short Comment (SC1) · 19 Oct 2017

The average GEM re-emission fluxes measured by Kamp and co-authors are a factor of 10 to 1000 higher than fluxes measured by other studies (Table 1). Maximum fluxes of 190ng m-2 min-1 were reported. On April 30 GEM re-emission fluxes were larger than 40 ng m-2 min-1 for a period of at least 4 hours, resulting in a conservative estimated re-emission of 10 ug Hg. At a comparable Arctic coastline site affected by AMDE's total Hg Pools of a maximum of 0.5 ug m-2 were reported for barrow in Barrow, Ak which is a factor of 20 lower than the re-emitted Hg reported by Kamp et al. (Johnson, K. P., et al. (2008), J. Geophys. Res., 113, D17304, doi:10.1029/2008JD009893) I

would like to suggest to the authors to perform a feasibility study and integrate the total amount of Hg that was re-emitted during the strong re-emission event on April 30 and compare it to 1) typical snow Hg pools measured at the study site and 2) the height of the atmospheric column that would need to be depleted of Hg during an AMDE to achieve such high Hg snow pools.

---

## Referee Comment (RC2) · Anonymous Referee #3 · 25 Oct 2017

[Summary]

This study reports the measurements of gaseous elemental mercury (GEM) fluxes over the snow cover at Station Nord, Greenland, from late April to mid-May. The authors employed a relaxed eddy accumulation (REA) technique, which is new for its application to the determination of GEM fluxes in the snow-covered polar region. They observed occasional, large emissions of elemental mercury from the snow surface, which is speculated to have resulted from the prior deposition of gaseous oxidized mercury (GOM). I believe that the data presented here are valuable additions to the literature obtained by a micro-meteorological technique rarely employed to date for the

measurements of mercury fluxes over the snow cover. In particular, it is found that the mercury (re-)emissions from the Arctic snow surface could take place, at least occasionally, at substantially greater rates than reported in earlier field studies using other types of methodology. The authors convey well the description of their methodology in sufficient details and yet quite concisely so that readers can understand the scientific background of the method, the instrumental configuration in the field and the data screening criteria. On the other hand, the quality of presenting results and discussion needs some significant improvement in both the presented contents and language to make this paper more compelling than currently is, hence the rating of "fair" for "Scientific Quality" and "Presentation Quality". I recommend major revisions before the present work is considered for final publication.

[Major comments]

1. Section 3 (Results and discussion) appears to need a thorough re-writing, as there are vague statements and unclear logical flows quite often. In addition, some of the speculative statements are given to sound as if they were evidenced in the present study. Since there were no measurements of GOM conducted in this study, any discussions related to the involvement of bromine chemistry leading to the oxidation of GEM to GOM and its temperature dependence as well as the deposition of GOM as a source of GEM re-emitted from the surface snow are all no more than speculations based on prior knowledge of AMDEs. In this regard, the illustration in figure 8 is not really based on evidence from this study and therefore should be dropped. On the same ground, the occurrence of very shallow surface inversions that did not reach the height of instruments for the field data acquisition and its connection to the temporal variations of measured GEM concentrations and fluxes remain speculative, although quite plausible. In my opinion, the sentences in section 3 must be composed more clearly to distinguish between facts and speculations. I also suggest the authors to cut back on the amount of speculative discussions and to increase fact-based and/or quantitative arguments such as those suggested below.

[Figure]

2. One of the major findings reported in this study is that the magnitude and rate of GEM (re-)remissions from the springtime Arctic snow surface can be much larger than previously reported. In addition to difference in time and location of measurements from earlier studies, the flux estimation technique employed here is different and more sophisticated. To increase the value of the present work, the authors could elaborate more on discussions related to the technical advantage of the REA method over the aerodynamic gradient method (AGM) and whether the two methods can result in significantly different data approval/rejection characteristics, for example, under windy conditions such as during April 26-30. I suppose that small vertical concentration gradients under the enhanced turbulent mixing render the AGM more or less inaccurate as you approach the limit of instrument accuracy. Is it not possible to perform some quantitative micro-meteorological (mathematical) arguments by estimating vertical scalar (tracer) diffusion coefficients and the GEM concentration gradients between the two hypothetical heights above the ground using the present field data, by which the authors could demonstrate the potential advantage of the REA method under windy conditions? I mean, hypothetically. In other words, had the authors derived the GEM fluxes by AGM instead of (or in addition to) REA, could the results have been largely the same? Such arguments could also help decrease the amount of speculative and qualitative statements in section 3.

3. Given the orders of magnitude greater re-emission fluxes of GEM than reported previously, the authors should provide a more detailed description of synoptic meteorology during April 26-30 when the episodes of large GEM emissions occurred. Showing a synoptic weather map or two if available and briefly explaining synoptic conditions around the study site (e.g., passage of cyclones) would be great; even greater if such weather maps could be associated with the time series of meteorological data presented in figure 6 and backward trajectories presented in figure 3. The passage of cyclones could also enhance bromine chemistry and hence the production of GOM in the atmospheric boundary layer, potentially serving as a fresh source of oxidized mercury in the surface snow (e.g., Zhao et al., ACPD, 2017, https://doi.org/10.5194/acp-

2017-427; Toyota et al., ACP, 2014, https://doi.org/10.5194/acp-14-4135-2014); it may be interesting to check with satellite BrO data if the authors can manage within the time frame of manuscript revision.

[Minor comments]

1. Equation (2): Something seems to be missing in these equations; as currently formulated, $C_{up} = C_{zero\,air}$ and $C_{down} = C_{zero,}$. I guess $C_{up}$ and $C_{down}$ on RHS must be multiplied by $\alpha_{up}$ and $\alpha_{down}$, respectively. Please double check.

2. Throughout section 3, the authors use the term "GOM" to refer to oxidized mercury retained in the snow after its deposition from the atmosphere. It should have been referred to differently, perhaps simply by "oxidized mercury".

[Technical suggestions]

P1, L29: inexplicit -> uncertain

P1, L29: relaxation -> residence

P2, L1: the sea -> seawater

P3, L30: backwards -> backward

P4, L9: the vertical turbulent flux of transported quantity is

P4, L23: must be larger than this threshold FOR AIR SAMPLES to be collected.

P9, L3: strongLY stable

P10, L6: extant -> of mercury chemistry and transport dynamics

P16-17, Figure 3: Add (a), (b), (c) and (d) on top of the trajectories maps.

P18, Figure 4: Add (a) and (b) on top of the trajectory frequency maps.

P21, Table 1: The range of GEM fluxes reported in the present study should be -8.0 to 190 ng m$^{-2}$ min$^{-1}$. Also, it seems useful to include the time (season) of data collection

for each study.

---

## Author Comment (AC1) · 20 Dec 2017

General comments: The authors present a first paper using the relaxed eddy accumulation (REA) method to measure flux over an Arctic snow covered surface, providing useful results for those looking to undertake similar Arctic mercury flux studies in the future. However, thorough editing of the text for grammar and clarity should be undertaken; in many areas, sentence fragments are present (see technical corrections for

examples), and this should be corrected prior to publication, to increase the comprehensibility of the text. In addition, I would also caution that the authors appear to be extrapolating their conclusions from a relatively limited data set, and many of the conclusions drawn from their results do not appear to be adequately supported by the presented results. In their Fig. 7a, GEM fluxes appear to be predominantly near 0, which means that any derivation of relationships based on this data set must be approached with caution, and broad sweeping statements about the importance of various factors should be avoided (see specific comments below).

Specific comments: 1. Use of the REA method: The manuscript presents a study using a technique (REA) which had not previously been used for this type of Arctic GEM flux work; however, the authors fail to explain why they have chosen this technique. In a "first use" study such as this it is important to explicitly outline the benefits of this technique over more traditionally used or other available methods, as well as the potential short-comings. In its present state, it is unclear why this work using the REA method is a benefit to the Arctic Hg flux literature, or why one may choose to do a REA based Hg flux study rather than use a chamber or AGM type method. »»» »»» We agree that a more detailed argument for choice of measurement method is needed. Therefore following text has been inserted into the manuscript at page 3 line 5: Chamber methods are attractive methods for measuring fluxes because of their low cost and simplicity but suffers from a number of weaknesses. They only capture the flux over a small area, the chamber affects the surface over which the measurement is taken and they can modify physical properties such as light and temperature (Bowling et al., 1998, Fowler et al., 2001). This implies that the measured flux will differ from the natural flux. The AGM is not altering the surface; however, it requires a homogeneous surface several hundred meters upstream the measurement site. Furthermore, it is assumed that the vertical profile is only a product of the vertical turbulent transport; nevertheless fast chemical reactions can affect the profile. The most direct flux measurement technique is the eddy covariance (EC) technique (Buzorius et al., 1998) but close to the surface this technique only works for fast responding monitors (sampling frequency >5 Hz), which

is not available for Hg. Therefore, we chose to employ the relaxed eddy accumulation (REA) method (Businger and Oncley, 1990) which is based on EC and the method does not affect the surface. Oncley et al. (1993) reported results with agreement within 20% for EC and REA and a study by Hensen et al. (1996) shows agreement between EC and REA within 10%, a difference that is reported not to be significant because the main error for REA is the determination of the concentration difference.

2. Exclusion of data (primarily CO2 flux to determine b section, and one Results reference): It is not immediately clear how the limits were chosen for various parameters to allow for inclusion or exclusion of data (in "CO2 flux to determine b" section). In total, the authors state that they excluded 74% of the collected data, which seems rather extreme, especially lacking adequate justification for the exclusion criteria. The authors state that they have estimated an uncertainty for b («10%), and state that the uncertainty is then assumed to be insignificant, but provide no methodology for that estimation, or justification for the assumption of insignificance. In this section, the authors also include a "z/L" criteria as a means by which data were included/excluded from the presented results, but this "z/L" is neither defined, nor described. Various parameters are given, however the choice to discard data should be more thoroughly explained and justified, especially when it means that very little of the originally collected data is included and used to derive the final relationships presented in this work. In the Results and Discussion (pg. 7, line 18 – 19), the authors propose a means by which some data may be falsely interpreted as outliers, but it is unclear whether this was a problem in the presented data set, and, if so, how was this dealt with? »»»> »»»> We agree it is unclear how the various parameters to exclude data is chosen. A more detailed explanation for the choice of the limits of b and the uncertainty is needed, as well as an explanation of z/L. Furthermore we have made further studies of the uncertainty of b and have adjusted our estimates and the discussion of uncertainty. Therefor the section 2.5 has been changed to the following text:

[revised manuscript text omitted]

3. Uncertainty in presented data (pg. 6, lines 12 – 14): The authors provide an estimate of the uncertainty "of the two concentration determinations", but it is unclear what precisely they mean by "the two concentration determinations"; is this the uncertainty of Hg flux based on the instrumental detection limit of 0.1 ng/m3? If so, this needs to be clarified (and units should be included with the appropriate numbers). In addition, it is stated that the uncertainty of the flux (which is given as 0.14) becomes 28% at the 95% confidence interval (unclear how this was derived), and then in the following sentence the authors give a GEM uncertainty of 10% above 0.5 ng/m3, but this is attributed to another study, so it is unclear how this fits in with any of their data, or uncertainty in their data, and precisely what the uncertainty on any of the provided results would be. In addition, no factors besides flux and b are given any consideration with respect to uncertainty, which makes it impossible to know how robust any of the presented results/relationships are. »»» »»» The section of uncertainty has been revised (see text under section 2 above)

4. Results and Discussion: I believe that this section of the manuscript should be thoroughly reviewed by the authors, and revised prior to publication, as many conclusions/relationships appear to have been drawn from GEM flux results that, as presented, typically seem to be approximately zero (Fig. 7a). In instances where the authors are calling on certain events (eg/increases in flux) these events should be explicitly stated, for clarity. »»> »»> We have changed figure 7, so it becomes more clear. We have added numbers to Fig. 7 to refer to the events and refers to the numbers in the discussion. Furthermore, we have added an extra figure showing the relation between the fluxes and temperature. The whole section has been thoroughly revised and rewritten.

5. The importance of heat/temperature changes: On pg. 2 (lines 30 – 31) the authors propose that a correlation between solar radiation induced heat flux/temp change and GEM production/flux exists, due to the observed relationship between solar radiation and GEM concentration increases in surface snow, and then again in the results/discussion (pg. 8, lines 9 – 12) they propose a temperature dependence in their observed results, and in the conclusions (pg. 9, lines 29 – 31) state that their data support the hypothesis that heating of the surface is influencing GEM formation/emission; however, this ignores the generally accepted idea that GEM production in snow (and many other media) is the result of photochemical reduction of Hg2+ to produce Hg0. It is possible that this process (Hg2+ reduction or subsequent Hg0 emission) from snow may be influenced by heat, but the primary reason for an increase in Hg0 production with increased solar radiation intensity will most likely be due to increases in the extent of Hg2+ photoreduction, not as a result of heat induced processes in the snowpack. In addition, in the results/discussion section, the authors state "After deposition, we speculate that GOM is reduced photolytically to GEM." (pg. 6, line 23 – 24), but then later state "GOM is reduced at the snow surface when temperature increases (Ferrari et al., 2008)." (pg. 6, line 26). GOM in the snow will most likely be reduced by some manner of photochemical reduction reaction (whether this be photolytic, or as a result of some other photochemically driven process), and while temperature may influence the extent of this reaction, or movement of GEM from the snowpack, radiation (sunlight) is required for the reaction to proceed. As it is used in the aforementioned statement, the Ferrari et al. (2008) reference is somewhat misleading, and this should be revised. In addition, the authors are using air temperature rather than the snowpack surface temperature to derive relationships with flux, and these will differ. Since the surface snow temperature may be significantly different than the ambient air temperature, it is difficult to make compelling conclusions about temperature dependence on snowpack Hg flux without these snowpack temperatures. »» »» We are aware of the reduction of GOM and possible subsequent emission of GEM and this was also our initial hypothesis. We investigated the relation between solar radiation and GEM emission but found no clear relation. However we found a possible correlation between temperature and GEM emission, which we think could be an important information to the scientific society measuring Hg fluxes in the Arctic, and we did not think it would be right to ignore this. We have inserted a figure showing the relation between GEM flux and radiation and the GEM flux and temperature, stating we are aware that this is the temperature in the atmosphere, since we unfortunately did not measure the snow temperature. We have changed the text in the discussion so it becomes more clear that the relation between GEM flux and temperature as well as radiation was investigated. Also we now refer more clear to other studies which also found a (reduction/emission???) temperature relation. Furthermore, we have changed the text on page 2 line 30-31 to:

This is most likely due to photoreduction of GOM and subsequent emission of GEM; however, it is also possible that a correlation between solar radiation-induced parameters such as heat flux or temperature change and GEM fluxes exists, making it relevant to look into temperature and heat flux as well as radiation in relation to GEM flux.

In addition, the authors state that "The highest temperatures were found during events

with the largest emissions..." (pg. 7, line 32 – 33), but these events have not been explicitly pointed out to the reader using the dates provided in the figures. From the data, I can see three easily observable GEM flux increase events (April 27, 28, 30); however, in one of these three easily observable emission events (April 28) increasing GEM fluxes occur before temperatures begin to increase, and, at least at the beginning of the GEM flux increase event, temperature is actually decreasing, and this is counter to what the authors have stated in their text. As a result, the text should be revised to explain this divergence from what they are stating to be typical behaviour (if more than the three observable events are being invoked). »» »» We have added a figure (Fig. 9) showing the relation between GEM flux and atmospheric temperature. We found no clear correlation between GEM flux and radiation, but we are aware of the relation. It is true that the relation between GEM and temperature and latent and sensible heat flux is not straightforward. This is because other parameters are also affecting the size of the flux. This has been written more clearly in the text now.

The authors use an instance of an upward latent heat flux (presumably indicated as a positive number in Fig. 7e?) on April 27 which coincided with a GEM emission event as evidence to support their GEM flux temperature/water dependence hypothesis; however, this increase in LH flux appears to be quite small, and while it does coincide with an increase in GEM flux, other increases in LH flux which appear to be comparable in magnitude (eg/April 25 – 26) did not result in increased GEM fluxes. Further, the second observed increase in GEM flux (April 28) appears to have occurred with no significant change in LH flux, and the largest observed increase in GEM flux (April 30) occurred with a LH flux decrease (negative value) that was much more significant than any of the LH flux increases observed in the data set. At best, it would appear that the data presented by this work appears to neither support nor dispute the hypothesis of temperature dependence of Hg flux based on LH flux information, but it supplies no compelling evidence to support it. If the latent heat flux argument is to be included, other incidents which were counter to the authors' hypothesis must also be discussed in the text, and adequate reasoning provided as to why these do not give evidence to

disprove the proposed hypothesis. »»> »»> The event on April 30 is an extreme event caused by a strong change in the meteorological conditions (possible a front passing) and as we have pointed out in the text this should not be a part of the general analyses. We have tried to make it more clearly in the text. It is true that many other parameters are influencing the flux and concentration of GEM. We have tried to make this more clear in the result and discussion.

6. The effects of wind speed on GEM flux: The authors state that all large emission events occurred when wind speeds increased (pg. 7 line 10), but it is unclear how wind speed effects and temperature effects are distinguished, or whether the proposed temperature effect on flux (see above) is simply the result of greater wind speeds appearing to coincide with increases in temperature (Fig. 6). With so many variables changing (potentially independently) at the same time, it is not possible to tease apart the relative importance of these on GEM flux by visual observation alone, which is how the proposed relationships appear to have been derived. »» »» It is true many parameters are affecting the flux and we have tried to show that especially for GEM the temperature is special since we don't see the same relation between $CO_2$ and temperature, however the relation between $CO_2$ and wind speed is the same as for GEM and wind speed.

7. Conclusions based on flux data trends: The authors attempt to determine relationships and draw conclusions regarding the effects of various factors on GEM flux; however, in looking at the data in Fig. 7a, it appears that in most instances, GEM flux is at or very near 0 ng/m3; while this may be a function of the scale being used, there simply does not appear to be adequate data to support strong conclusions, with the data presented, especially with no inclusion of the uncertainty on these data. Further, if the authors plan to make conclusions regarding the dependence of flux on the various factors that they measured, it would be very helpful to have some manner of statistical test to back up these claims, as without them, the invoked trends are neither clear nor compelling. For example, the authors state that depletion events on April 23 – 25 and

May 2 – 5 are followed by GEM emissions; however, while this may be supported by the April 27 increase in GEM flux that is visible in the results, there does not appear to be a significant increase in GEM flux following May 5, until the small (almost apparently negligible) increase in flux on May 7, when GEM concentrations are higher again. With the data as it has been presented, this does not appear to support GEM emission post-AMDE as the authors have proposed (pg. 6 line 21). »»» »»» We have changed part of the conclusion and made it more subtle: The results of this study supports to some extent the general understanding of the AMDE mechanisms where GEM oxidation is followed by deposition of GOM, which is partly reduced to GEM and reemitted into the atmosphere. Furthermore, the data indicates that heating of the snow surface influences formation of GEM and reemission of GEM.

8. Correlation between CO2 and GEM: The authors state that there is a correlation between CO2 and GEM, but offer no methods used to determine this. Was this decided based on mathematical/statistical analysis? Simple observation? Is this GEM concentration, or GEM flux? Based on a quick visual inspection of the results in Fig. 7a/b and 7c, there does not appear to be a compelling case for simple visual observation of such a trend. If this conclusion is to be included in the paper, there should be a more thorough investigation of the claim, or the methods used to draw this conclusion should be explicitly stated, at the very least.

»» »» The relation between CO2 and GEM was suggested by the editor. We have now added figure 10, which shows the co2 and GEM flux in relation to wind speed and it by visual observation we see an anti-correlation between the two fluxes. It is explained more careful in the text

9. Comparison of study results with literature results (pg. 7 lines 19 – 24, pg. 8 line 15 – pg. 9 line 21): Overall, the authors' discussion of their results as compared to other literature results is somewhat difficult to follow, and should be revised for clarity. This portion of the manuscript would benefit from the inclusion of more concrete results and explicit discussion (eg/ pg. 7 line 23 – 24: how can your results being in opposition to

those from the Osterwalder study be explained by the difference in location? How is the GEM dynamic different in these studies? eg/pg. 8 line 21: what was your net emission, exactly? What were the values found by the other works that are referenced?), and from complete discussion of one topic before moving on to another to improve flow and comprehensibility (eg/ stability conditions are discussed in more than one place). Overall, as a discussion paper, comparison with other studies should be much clearer, allowing the reader to easily place the present study with those already existing in the literature (does it agree with other studies using similar method or not, and why?), and at present this is not the case. It is further unclear, in some instances, how certain discussions relate to the present study. For example, when discussing stable conditions/GEM build-up (pg. 9, lines 3 – 10), the authors state that strong stratification with a build-up of GEM near the surface will result in violation of a basic assumption for the flux gradient method; was this phenomenon expected in your study, and if so, how did you deal with it? If this violation of a basic assumption for the flux gradient method was not observed in your study, why have you included it here? The discussion section (and comparison to other studies) might also be easier to follow if a better introduction to the chosen technique was included (see specific comment #1). »»> »»> We agree that this could be more clear and the discussion has in general been cleaned so it is follows the recommendation of the reviewer.

The phrase opposite concerning Oswalds observations is changed to: "On the other hand, Osterwalder et al. (2016) observed emission during unstable conditions, a small deposition during stable conditions and deposition during neutral conditions."

Regarding the strong stratification and assumptions for different flux measurement techniques, we expect the reader to be familiar with the basics of the different techniques, but see answer to comment 1 where justification for the method is added. To the introduction the following has been added: "Furthermore, strong stratification violates the assumption of gradient measurements, thus REA is in our opinion the best possible option to measure GEM flux."

10. Results figures/tables: For the results figures (Fig. 6 and 7), the markers/scales chosen make it almost impossible to see differences in the data with time, except where those differences are very large. Since the authors are attempting to use such differences in various measurements over time to derive information regarding factors influencing GEM flux, it is imperative that the reader be able to see the differences the authors appear to be speaking of, and at present, this is not true in most cases. In addition, where uncertainty is known (eg/flux uncertainty = 28%, as stated in the manuscript) error bars should be provided for the data in these figures, as it is unclear whether changes (eg/in flux over time) might be statistically significant, or not. Also, certain events (eg/ AMDEs as in pg. 9 line 26 – 27: "...during which several AMDEs were observed.") should be explicitly marked on your data, or the dates you are proposing they have occurred should be present in the text and/or figure captions. »»> »»> We agree that this should be more clear. This has been revised accordingly (see previous comments and text).

In the summary table (Table 1, pg 21) the authors give a flux range of 8 – 190 ng m-2 min-1 for their data set; however, in looking at the results in Fig. 7a, it is apparent that there were some incidents of negative (depositional) flux (April 27), and there are many instances where the flux appears to be zero. As a result, it appears that the flux range given in the table is either incorrect, or some values were excluded, and if they were excluded, a reason should be given for this, as the table is not particularly informative without it. »»> »»> There is an error in table 1. The range should be -8.1 to 179.2 ng m-2 min-1. This is now corrected both in the main text and the table.

Technical corrections: Pg. 1 line 27 – pg. 2 line 2: These two sentences appear to be contradictory, and it is unclear what the authors are attempting to inform the reader of re: atmospheric lifetime of GEM with the given text. »»> This sentence is now removed since it confuses the reader instead of enlighten.

Pg. 1 line 29: What do you mean by "...the relaxation time of mercury in the atmosphere..."? Please clarify/revise. »»> This sentence is removed as part of the

sentence above.

Pg. 2 line 4 – 5: Sentence fragments; consider adding fragment "These atmospheric mercury depletion events. . ." to the previous sentence. »»> Changed as suggested, and made into one sentence.

Pg. 2 line 8 – 9: "Thus, this is a human health. . ." sentence fragment. »»> Changed to one sentence.

Pg. 2 line 13: Consider revising ". . .concentration decreases during. . ." to ". . .concentration decreases due to. . ." »»> Changed as suggested.

Pg. 2 line 18 – 19: "Vertical gradient. . ." is repeat of the information presented in previous sentence, consider revising. »»> We agree, and have removed the sentence

Pg. 2 line 22 – 24: "This is likely due to. . ." sentence fragment, consider adding to the previous sentence. »»> Changed as suggested.

Pg. 2 line 25: affect, not affects. »»> Changed as suggested.

Pg. 2 line 30 – 32: "Thus it is likely that a correlation. . ." this statement is somewhat misleading, as increases in solar radiation lead to increases in GEM production/emission as a result of an increase in Hg2+ reduction, which may have nothing to do with heat! »»> This statement is changed to: "This is most likely due to photoreduction of GOM and subsequent emission of GEM; however, it is also possible that a correlation between solar radiation-induced parameters such as heat flux or temperature change and GEM fluxes exists, making it relevant to look into temperature and heat flux as well as radiation in relation to GEM flux."

Pg. 5 line 19: The detection limit you have given is the literature value for the instrumental detection limit, which may be significantly different than your method detection limit. Consider revising to be the method detection limit. »»> It is specified, and a section on errors has been added, see above.

[Figure]

Pg. 5 line 25: "EC" is used without definition in the text (provided in the abstract, but should also be included on first use in the main body of the manuscript). »»> Changed as suggested.

Pg. 5 line 26 – 28: "The flux of CO2…" Sentence is confusing, please revise. »»> The sentence has been rewritten.

Pg. 6 line 2: Was b derived based on T (presumably temperature?)? If so, this method has not been described. »»> A sentence has been added.

Pg. 6 line 7: What is z/L? This is not defined. »»> A sentence is added.

Pg. 6 line 26 – 27: "This leads to increased…" Sentence fragment, please revise. »»> Rephrased as part of revision of section 3.

Pg. 6 line 31 – pg.7 line 2: "We observed a clear diurnal pattern…" Did you measure incident solar radiation intensity? If so, this data should be included, and if not, how have you arrived at this conclusion/the timing of max and min sunlight? »»> A reference to figure 9a is added, which shows no correlation between the flux and solar radiation. We think it is redundant to show a graph with the diurnal pattern, as there is no correlation.

Pg. 7 line 15 – 18: "This could have occurred…" Sentence fragment, please revise. »»> Rephrased as part of revision of section 3.

Pg. 7 line 23 – 24: "These differences can be explained…" Sentence fragment, please revise. »»> It is specified to differences in emission during different stabilities.

Pg. 8 line 3: "At low temperature (< -20 C) the flux of GEM was near zero." GEM flux was also near zero in many cases when the temperature was > -20 C; this should be discussed. »»> It is true that the GEM flux was near zero in many cases, so the sentence is rephrased as there are only fluxes close to zero < -20 C with a reference to fig 9b.

Pg. 8 line 24 – 25: "This could be due to higher wind speeds..." Sentence fragment, please revise. »»> The two sentences are combined.

Pg. 8 line 31 – 34: The two sentences about the Manca et al. (2013) study should be combined, as the second is mostly redundant. »»> Rephrased as part of revision of section 3.

Pg. 9 line 14 – 15: "Differences in locations between research sites..." Please state why this is important and/or how this is expected to influence your study with relation to others, as there is no context for this statement at present. »»> The sentence is coupled to the next and rephrased to specify why the differences are important.

Pg. 9 line 15: "Important parameters are..." Sentence fragment, please revise. »»> Changed with the previous comment.

Pg. 9 line 29 – 31: I don't believe you have provided a compelling case to support the given hypothesis, as the manuscript stands. »»> We hope the changes made in the manuscript are sufficient to support this sentence.

Pg. 14 Fig. 1: It's a little bit difficult to find the yellow dot in your figure, consider giving this dot a dark coloured outline to increase visibility where it overlies the white page. »»> We think that the position is already pointed out in the caption and by visual contrast, so nothing has been changed here.

Pg. 15 Fig. 2 caption: Should be "indicates", not "indicate". »»> Changed as suggested.

Pg. 16 Fig. 3 caption: Should be "shows" not "show". »»> Changed as suggested.

Pg. 19/20 figure labels: authors are not consistent with the way they are writing units in the text vs. the figures (eg/ ng/m3 in text vs. ng m-3 in figures). »»> All cases have been changed, expect z/L that is a normal term in meteorology.

Pg. 20 Fig. caption: "mol" not "mole". »»> Changed as suggested.

Pg. 21 Table title: Should be "Summary table of…" rather than "Summary table over…" »»> Changed as suggested.

Please also note the supplement to this comment:
https://www.atmos-chem-phys-discuss.net/acp-2017-518/acp-2017-518-AC1-supplement.pdf

---

## Author Comment (AC2) · 20 Dec 2017

[Summary] This study reports the measurements of gaseous elemental mercury (GEM) fluxes over the snow cover at Station Nord, Greenland, from late April to mid-May. The authors employed a relaxed eddy accumulation (REA) technique, which is new for its application to the determination of GEM fluxes in the snow-covered polar region. They observed occasional, large emissions of elemental mercury from the snow

surface, which is speculated to have resulted from the prior deposition of gaseous oxidized mercury (GOM). I believe that the data presented here are valuable additions to the literature obtained by a micro-meteorological technique rarely employed to date for the measurements of mercury fluxes over the snow cover. In particular, it is found that the mercury (re-)emissions from the Arctic snow surface could take place, at least occasionally, at substantially greater rates than reported in earlier field studies using other types of methodology. The authors convey well the description of their methodology in sufficient details and yet quite concisely so that readers can understand the scientific background of the method, the instrumental configuration in the field and the data screening criteria. On the other hand, the quality of presenting results and discussion needs some significant improvement in both the presented contents and language to make this paper more compelling than currently is, hence the rating of "fair" for "Scientific Quality" and "Presentation Quality". I recommend major revisions before the present work is considered for final publication.

[Major comments]

1. Section 3 (Results and discussion) appears to need a thorough re-writing, as there are vague statements and unclear logical flows quite often. In addition, some of the speculative statements are given to sound as if they were evidenced in the present study. Since there were no measurements of GOM conducted in this study, any discussions related to the involvement of bromine chemistry leading to the oxidation of GEM to GOM and its temperature dependence as well as the deposition of GOM as a source of GEM re-emitted from the surface snow are all no more than speculations based on prior knowledge of AMDEs. In this regard, the illustration in figure 8 is not really based on evidence from this study and therefore should be dropped. On the same ground, the occurrence of very shallow surface inversions that did not reach the height of instruments for the field data acquisition and its connection to the temporal variations of measured GEM concentrations and fluxes remain speculative, although quite plausible. In my opinion, the sentences in section 3 must be composed more

clearly to distinguish between facts and speculations. I also suggest the authors to cut back on the amount of speculative discussions and to increase fact-based and/or quantitative arguments such as those suggested below.

»»> »»> Figure 8 has been removed. We have changed figure 7, so it becomes more clear. We have added numbers to Fig. 7 to refer to the events and refers to the numbers in the discussion. Furthermore, we have added an extra figure showing the relation between the fluxes and temperature. The whole section has been thoroughly revised and rewritten.

2. One of the major findings reported in this study is that the magnitude and rate of GEM (re-)remissions from the springtime Arctic snow surface can be much larger than previously reported. In addition to difference in time and location of measurements from earlier studies, the flux estimation technique employed here is different and more sophisticated. To increase the value of the present work, the authors could elaborate more on discussions related to the technical advantage of the REA method over the aerodynamic gradient method (AGM) and whether the two methods can result in significantly different data approval/rejection characteristics, for example, under windy conditions such as during April 26-30. I suppose that small vertical concentration gradients under the enhanced turbulent mixing render the AGM more or less inaccurate as you approach the limit of instrument accuracy. Is it not possible to perform some quantitative micro-meteorological (mathematical) arguments by estimating vertical scalar (tracer) diffusion coefficients and the GEM concentration gradients between the two hypothetical heights above the ground using the present field data, by which the authors could demonstrate the potential advantage of the REA method under windy conditions? I mean, hypothetically. In other words, had the authors derived the GEM fluxes by AGM instead of (or in addition to) REA, could the results have been largely the same? Such arguments could also help decrease the amount of speculative and qualitative statements in section 3.

»»> »»> We agree that a more detailed argument for choice of measurement method

is needed. Therefore following text has been inserted into the manuscript at page 3 line 5: "Chamber methods are attractive methods for measuring fluxes because of their low cost and simplicity but suffers from a number of weaknesses. They only capture the flux over a small area, the chamber affects the surface over which the measurement is taken and they can modify physical properties such as light and temperature (Bowling et al., 1998, Fowler et al., 2001). This implies that the measured flux will differ from the natural flux. The AGM is not altering the surface; however, it requires a homogeneous surface several hundred meters upstream the measurement site. Furthermore, it is assumed that the vertical profile is only a product of the vertical turbulent transport; nevertheless fast chemical reactions can affect the profile. The most direct flux measurement technique is the eddy covariance (EC) technique (Buzorius et al., 1998) but close to the surface this technique only works for fast responding monitors (sampling frequency >5 Hz), which is not available for Hg. Therefore, we chose to employ the relaxed eddy accumulation (REA) method (Businger and Oncley, 1990) which is based on EC and the method does not affect the surface. Oncley et al. (1993) reported results with agreement within 20% for EC and REA and a study by Hensen et al. (1996) shows agreement between EC and REA within 10%, a difference that is reported not to be significant because the main error for REA is the determination of the concentration difference."

3. Given the orders of magnitude greater re-emission fluxes of GEM than reported previously, the authors should provide a more detailed description of synoptic meteorology during April 26-30 when the episodes of large GEM emissions occurred. Showing a synoptic weather map or two if available and briefly explaining synoptic conditions around the study site (e.g., passage of cyclones) would be great; even greater if such weather maps could be associated with the time series of meteorological data presented in figure 6 and backward trajectories presented in figure 3. The passage of cyclones could also enhance bromine chemistry and hence the production of GOM in the atmospheric boundary layer, potentially serving as a fresh source of oxidized mercury in the surface snow (e.g., Zhao et al., ACPD, 2017, https://doi.org/10.5194/acp-

2017-427; Toyota et al., ACP, 2014, https://doi.org/10.5194/acp-14-4135-2014); it may be interesting to check with satellite BrO data if the authors can manage within the time frame of manuscript revision.

»»> »»> The event on April 30 is an extreme event caused by a strong change in the meteorological conditions (possible a front passing) and as we have pointed out in the text this should not be a part of the general analyses. We have tried to make it more clearly in the text, thus we do not thing synoptic weather maps for this period is relevant.

[Minor comments]

1. Equation (2): Something seems to be missing in these equations; as currently formulated, Cup = Czero air and Cdown = Czero;. I guess Cup and Cdown on RHS must be multiplied by _up and _down, respectively. Please double check. »»> The equation is as it should be.

2. Throughout section 3, the authors use the term "GOM" to refer to oxidized mercury retained in the snow after its deposition from the atmosphere. It should have been referred to differently, perhaps simply by "oxidized mercury". »»> We agree, and have changed the use of GOM to oxidized mercury when it is retained in the snow.

[Technical suggestions] P1, L29: inexplicit -> uncertain »»> Sentence removed

P1, L29: relaxation -> residence »»> Sentence removed

P2, L1: the sea -> seawater »»> Changed as suggested.

P3, L30: backwards -> backward »»> Changed as suggested.

P4, L9: the vertical turbulent flux of transported quantity is »»> Changed as suggested.

P4, L23: must be larger than this threshold FOR AIR SAMPLES to be collected. »»> Changed as suggested.

P9, L3: strongLY stable »»> Changed as suggested.

P10, L6: extant -> of mercury chemistry and transport dynamics »»> Changed as suggested.

P16-17, Figure 3: Add (a), (b), (c) and (d) on top of the trajectories maps. »»> Changed as suggested.

P18, Figure 4: Add (a) and (b) on top of the trajectory frequency maps. »»> Changed as suggested.

P21, Table 1: The range of GEM fluxes reported in the present study should be -8.0 to 190 ng m-2 min-1. Also, it seems useful to include the time (season) of data collection for each study. »»> There is an error in the table, which has been corrected.

Please also note the supplement to this comment:
https://www.atmos-chem-phys-discuss.net/acp-2017-518/acp-2017-518-AC2-supplement.pdf

---

## Author Comment (AC3) · 20 Dec 2017

M. Jiskra martinjiskra@gmail.com The average GEM re-emission fluxes measured by Kamp and co-authors are a factor of 10 to 1000 higher than fluxes measured by other studies (Table 1). Maximum fluxes of 190ng m-2 min-1 were reported. On April 30 GEM re-emission fluxes were larger than 40 ng m-2 min-1 for a period of at least 4 hours, resulting in a conservative estimated re-emission of 10 ug Hg. At a comparable Arctic coastline site affected by AMDE's total Hg Pools

of a maximum of 0.5 ug m-2 were reported for barrow in Barrow, Ak which is a factor of 20 lower than the re-emitted Hg reported by Kamp et al. (Johnson, K. P., et al. (2008), J. Geophys. Res., 113, D17304, doi:10.1029/2008JD009893) I would like to suggest to the authors to perform a feasibility study and integrate the total amount of Hg that was re-emitted during the strong re-emission event on April 30 and compare it to 1) typical snow Hg pools measured at the study site and 2) the height of the atmospheric column that would need to be depleted of Hg during an AMDE to achieve such high Hg snow pools. »»» »»»

First, we thank M. Jiskra for his comments. As stated already we measure significant higher fluxes of GEM at Villum Research Station (VRS) located at 81 degree north. Unfortunately, we are missing data for mixing height and the amount of mercury in the snow. We have only a few surface layer concentrations of elemental mercury in the snow, and we are not able to use the data to assess any mass balance for GEM. The measurement site in Johnson et al. (2008) and in the present study are both located at the cost, but due to the colder climate and nearby glacier, the conditions at VRS is very different from Barrow. Most of the year VRS is surrounded by ice and only in late July to September there is ice-free conditions around the peninsula, where VRS is located. During this period there is a strong flow of fresh water from the nearby ice sheet "Flade Isblink" (Bentzon et al. 2017, Scientific Reports | 7: 4941 | DOI:10.1038/s41598-017-05089-3). The upper 2 meters of water around the station is therefore fresh water. During spring, the weather is extreme with very low temperatures, stagnant wind and very low inversion height. First meteorological measurements from an 80 m mast often show inversion at a few 10's of meters. Therefore, GEM is emitted into a very shallow atmospheric layer, which could be the reason for the high fluxes observed. Unfortunately, these measurements were not carried out until 2017. Furthermore, a large part of the snow is drifting snow, so the origin of GEM in drifting snow is difficult to determine. In the article, we argue that chamber methods as conducted by Johnson et al. (2008) are very different from micrometeorological methods as REA. Enclosure methods as chambers can potentially change temperature, humidity,
radiation etc. (Fowler et al., 2001). Chambers cover a very limited area compared to micrometeorological methods. Due to the general differences in measurement methods we will not perform quantitative comparison between enclosure methods and micrometeorological. We argue that the event occurring on April 30 is most likely explained by sudden changes in meteorological conditions from a front passing and see this as an extreme case.

Please also note the supplement to this comment:
https://www.atmos-chem-phys-discuss.net/acp-2017-518/acp-2017-518-AC3-supplement.pdf

---

## Referee Report (RR1)

**Specific comments**

RE: temperature dependence of GEM production/emission:

The conclusion regarding temperature dependence of GEM production/emission is one of the major conclusions of the manuscript; however, I still do not believe compelling evidence has been provided to support this as a major conclusion of the work. It appears that the idea of higher temperatures resulting in higher GEM fluxes has been reached following visual observation of the data, but there is a great deal of scatter in the data, and this makes the conclusion potentially uncertain, and definitely less straightforward that has been presented in the manuscript. If this conclusion is to be included, it should certainly be qualified, and the shortcomings/uncertainty identified, as well as the process which lead to the identification of this "relationship".

The discussion of results is much more clear, in general; however, on pg. 8 line 15 - 16, the authors include the statement that: "*Low temperatures are required for the occurrence of AMDE (<-4 C)...*", and this condition represents the entirety of their data set. As a result, I do not believe this is a particularly compelling argument for the temperature relationship proposed in the manuscript.

In addition, the highest fluxes of GEM are certainly observed when temperatures are > -20 °C; however, this data is quite scattered, with many instances of 0 ng/m$^2$/h GEM fluxes being observed at T > -20 C. This would, then, not imply that the relationship between GEM flux and temperature is as simple as higher temperatures resulting in greater GEM production and emission, as may be implied from the conclusions as presented (ie/ pg 11, line 4 - 5 "Furthermore, the data indicate that that heating of the snow surface influences formation of GEM and reemission of GEM"). As a result, I would suggest further "softening" of this conclusion, and acknowledgement of the lack of a straighforward or definite relationship between these factors.

The statement about these GEM flux vs. temperature results that is presented in the abstract (pg 1, line 16 - 18: "*The measurements also indicate GEM emission is increasing with increasing temperature...*") is misleading, as it implies some manner of mathematical relationship could be derived, while the presented results simply show some instances of higher GEM fluxes when temperatures were above the ~ -20 °C threshold, coincident with many zero GEM flux measurements under those same temperature conditions. This statement should be revised alongside the conclusions.

Finally, when comparing the $CO_2$ fluxes vs. temperature in Fig. 9c, with the GEM fluxes vs. temperature in Fig. 9b, the authors state that there is no temperature dependence on $CO_2$ fluxes (pg. 8 line 12 – 14); however, simple observation of these two figures does not present a compelling case for the statement of a relationship in Fig. 9b vs. no relationship in 9c. If the authors are making the argument for relationships based strictly on visual observation, it appears that $CO_2$ fluxes may decrease with increasing temperature, where the highest depositional fluxes

of $CO_2$ appear to occur at $T > -20$ °C. If a more rigorous approach was taken to determine the presence of a relationship between temperature and GEM flux (vs. no relationship between $CO_2$ flux and temperature), this should be presented in the manuscript; however, if simple visual observation was employed, as appears to be the case with the manuscript in its current state, then the conclusions regarding the occurrence of a relationship in the GEM vs. temperature data (Fig. 9b) and no relationship in the $CO_2$ vs. temperature data (Fig. 9c) may be the result of observer bias, and should be reanalysed and/or not included as a major conclusion of the work.

---

## Author Response (AR2)

**Referee #2 comments**

**Specific comment:** RE: temperature dependence of GEM production/emission:
The conclusion regarding temperature dependence of GEM production/emission is one of the major conclusions of the manuscript; however, I still do not believe compelling evidence has been provided to support this as a major conclusion of the work. It appears that the idea of higher temperatures resulting in higher GEM fluxes has been reached following visual observation of the data, but there is a great deal of scatter in the data, and this makes the conclusion potentially uncertain, and definitely less straightforward that has been presented in the manuscript. If this conclusion is to be included, it should certainly be qualified, and the shortcomings/uncertainty identified, as well as the process which lead to the identification of this "relationship".

The discussion of results is much more clear, in general; however, on pg. 8 line 15 - 16, the authors include the statement that: "*Low temperatures are required for the occurrence of AMDE (<-4 C)...*", and this condition represents the entirety of their data set. As a result, I do not believe this is a particularly compelling argument for the temperature relationship proposed in the manuscript.

In addition, the highest fluxes of GEM are certainly observed when temperatures are > -20 °C; however, this data is quite scattered, with many instances of 0 ng/m2/h GEM fluxes being observed at T > -20 C. This would, then, not imply that the relationship between GEM flux and temperature is as simple as higher temperatures resulting in greater GEM production and emission, as may be implied from the conclusions as presented (ie/ pg 11, line 4 - 5 "Furthermore, the data indicate that that heating of the snow surface influences formation of GEM and reemission of GEM"). As a result, I would suggest further "softening" of this conclusion, and acknowledgement of the lack of a straighforward or definite relationship between these factors.

The statement about these GEM flux vs. temperature results that is presented in the abstract (pg 1, line 16 - 18: "*The measurements also indicate GEM emission is increasing with increasing temperature...*") is misleading, as it implies some manner of mathematical relationship could be derived, while the presented results simply show some instances of higher GEM fluxes when temperatures were above the ~ -20 °C threshold, coincident with many zero GEM flux measurements under those same temperature conditions. This statement should be revised alongside the conclusions.

Finally, when comparing the CO2 fluxes vs. temperature in Fig. 9c, with the GEM fluxes vs. temperature in Fig. 9b, the authors state that there is no temperature dependence on CO2 fluxes (pg. 8 line 12 – 14); however, simple observation of these two figures does not present a compelling case for the statement of a relationship in Fig. 9b vs. no relationship in 9c. If the authors are making the argument for relationships based strictly on visual observation, it appears that CO2 fluxes may decrease with increasing temperature, where the highest depositional fluxes of CO2 appear to occur at T > -20 °C. If a more rigorous approach was taken to determine the presence of a relationship between temperature and GEM flux (vs. no relationship between CO2 flux and temperature), this should be presented in the manuscript; however, if simple visual observation was employed, as appears to be the case with the manuscript in its current state, then the conclusions regarding the occurrence of a relationship in the GEM vs. temperature data (Fig. 9b) and no relationship in the CO2 vs. temperature data (Fig. 9c) may be the result of observer bias, and should be reanalysed and/or not included as a major conclusion of the work.

Response: We agree that our data do not show a clear relation between temperature and GEM emissions, and this is why we use words like "indicate" and "suggest" instead of "clearly show". However, we have now softened the conclusion further on GEM production/emission and temperature dependency.

In the abstract, the sentence: *"The measurements also indicate GEM emission is increasing with increasing temperature, supporting that surface heating controls GOM reduction in the surface layer of the snow."* Is replaced by: *Furthermore, observation of the relation between GEM fluxes and atmospheric temperature suggest that GEM emission partly could be affected by surface heating. However, it is also clear that the GEM emissions are affected by many parameters.*

In the conclusion, the sentence: *"Furthermore, the data indicate that heating of the snow surface influences formation of GEM and reemission of GEM".* Is now replaced by: *"Furthermore, the data show some relation between increase in upward GEM fluxes and increasing temperature and heating of the snow surface. However, the scatter on the flux data is large and the snow temperature is not measured in present study, thus further detailed studies to investigate this relation is needed."*

Furthermore we have extended the discussion on temperature in section 3 and referred to studies with a larger range of temperatures.

**Referee #3 comments**

**Major Comment 1:** One of the major GEM emission events, namely, "Event 1" on April 27, was accompanied by the downward sensible heat flux, indicating that temperature was lower in the snow surface than in the ambient air. Figure 6b shows the air temperatures varied between -10 and -15 degrees Celsius, so that the snow temperatures were certainly lower this temperature range during the event. The authors need to refer more explicitly to the range of temperatures reported in other studies where the temperature dependence was indicated in the in-snow reduction of oxidized Hg and the subsequent emission of GEM, i.e., whether or not previously reported temperature trends match the present findings such as those indicated in Figure 9b. Interestingly, the upward latent heat flux was observed during this event, suggesting that the sublimation (or evaporation) of water was probably occurring in the snow. One can also argue (hypothesize) that such loss of water mass from the snow grains could have facilitated the exposure of snow-deposited Hg to the air and could thus have played a role in the emission of GEM even if the efficiency of the chemical reduction of oxidized Hg is not correlated with the air exposure. Is there any relation between the measured latent heat and GEM fluxes?

Respond: To answer comment 1 we have now looked careful into the literature and we have now changed the text on page 8 and 9 to the following in order to discuss other researchers findings at low temperature in relation to our finding and. In the revised text we also discuss the radiation and latent heatflux in relation to this and find that the correlation between latent heat flux and GEM flux is not clear, but there is a possible relation between the sensible heat flux and GEM:

*At low temperature (< -20 ℃) only fluxes of GEM close to zero were present, see Figure 9b. Low temperatures are required for the occurrence of AMDE (< -4 ℃) (Lindberg et al., 2002;Skov et al., 2004), at which point GEM is oxidized to GOM. This indicates that GEM is so easily oxidized to GOM at lower temperatures (< -20 ℃) that GEM falls below the detection limit. This is consistent with the findings of Cole and Steffen (2010), Berg et al. (2013) and Cobett et al. (2007), which found the lowest concentrations of GEM (<0.5 ng/m3) at low temperatures (<-15) in spring after polar sunrise. Ozone and GEM depletion are correlated during AMDEs possibly due to reactions mainly with Br, and low temperatures favor the reaction between Br and GEM (Goodsite et al., 2004, 2012;Skov et al., 2004;Schroeder et al., 1998).*

*The measurements were started when depletion was already present and, as seen in Figure 8a and Figure 8b, depletion (low GEM concentration during April 23-25 (AMDE 1) and May 2-5 (AMDE 2)) was followed by GEM emission as observed by Brooks et al. (2006), supporting that GEM is reemitted after AMDEs. The results correspond to the general understanding that GEM is initially removed rapidly from the atmosphere. This removal is most likely due to photolytic oxidation to oxidized mercury, which, contrary to GEM, has a very low surface resistance (Skov et al., 2006) and thus deposits relatively quickly. It is generally accepted that GEM production in snow is the result of a photochemical reduction of oxidized mercury to produce GEM. Thus, we at first hypothesized that oxidized mercury is reduced photolytically to GEM in the surface snow followed by reemission. However, Ferrari et al. (2005) found that production of GEM is linked to the snow temperature and according to Steffen et al. (2015) and 2002, the photochemical reduction of oxidized mercury in snow – and thus the reemission of GEM – is temperature dependent. .Fain et al. (2013) concluded that temperature and solar radiation were the main environmental parameters controlling GEM production in snow and found increased GEM production in snow even at snow temperatures below -5C. Mann et al., 2015 found increased GEM flux from snow when the solar radiation and snow temperature increased; even at low air (-20 C) and low snow (-15 C) temperature. Furthermore they found in laboratory*

*studies that temperature influenced Hg photoreduction kinetics when the snow is approaching their melting point (>-2C), suggesting that temperature influences Hg photoreduction kinetics indirectly. Similarly, in the sub-arctic Dommergue et al. (2003) showed that melting snow emits more GEM than at lower snow temperatures. Therefore, an increase in atmospheric temperature and solar radiation increasing the snow temperature could lead to increased reemission of GEM causing the concentration of GEM in the atmosphere to increase.*

*In the present study, the largest emissions were found during events with the highest temperatures (temperatures > -15°C), as seen in Figures 9b. The same behavior is not found for $CO_2$ flux (Figure 9c), where fluxes measured from -20°C to -15°C have the same magnitude as fluxes measured from -15°C to -10°C. The mean fluxes of GEM and $CO_2$ for the temperature intervals 5°C to -10°C, -10°C to -15°C and >-20°C also show an increase in the emission of GEM at increasing temperature (See Table 2) but a less clear relation between $CO_2$ flux and temperature. Both GEM and $CO_2$ fluxes correlate with the wind speed (Figure 10) and stability, thus we argue that the temperature could be a possible driver for the GEM emissions presented here. Oxidized mercury species are water-soluble, hence it is assumed that reduction of deposited Hg takes place in the aqueous phase (Steffen et al., 2015), which is followed by emission of the more volatile GEM. It is possible that the temperature relation observed in present study is due to an increased water content in the snowpack. Heating of the surface (i.e. downward sensible heat flux) and upward latent heat flux (evaporation or sublimation) occurred on April 27 during the first larger GEM emission event (Event1), supporting the temperature- and water- dependency hypothesis. However, we found no strong relation between GEM flux and latent heat flux in general (See figure 11a), but we observed that high emission of GEM was in general associated with downward sensible heat fluxes (Fig 11b). A clear diurnal pattern for the radiation intensity was found, with the maximum at noon and the minimum at midnight, but these diurnal variations seem not to correlate with the GEM flux or concentration directly, see Figure 9a. Nevertheless, it is likely the snow is heated by the relatively strong solar radiation (> 400 wm-2) during the day and by the air, when this is warmer than the snow. Unfortunately, we did not measure temperature or humidity in the snow, to support the suggested relation between emission, snow melting and air temperature in our study.*

**Major Comment 2:** I am puzzled by the way the authors discuss what could have happened during and after "Event 2" from April 28 to April 29. Corresponding discussions in section 3 (starting from page 8, line 32) at first note the large GEM emission on April 28 when the stability changed from near-neutral to unstable conditions. The following sentence ("The day prior to this event…") is rather poorly constructed in conveying the information, but if I understand correctly, the authors are trying to indicate that the observed GEM concentrations were steady around 1 ng m-3 until the next day (April 29) when the stability changed again from stable to unstable conditions. I am basically fine with the authors' speculation in that the build-up of GEM in the stable surface boundary layer is often not detected by the field measurements simply because such build-up can occur in a way too shallow stable boundary layer, but the data are not presented in an efficient manner by scattering the relevant information between figures 6 and 8. It appears as if the authors were applying the same logic to what they see in "Event 3", but I am not sure. The authors should come up with a better way to present what are currently marked as "Event 2" and "Event 3" in the figures and corresponding discussions in section 3. In particular, the authors should revise the figures in that the GEM emission event currently marked as "Event 2" is presumably linked to the GEM concentration increase on the following day.

Respond: To answer comment 2 we have now edited the discussion carefully and made it more clear (the discussion on the issue of accumulated concentrations and venting of this was spread out randomly in the discussion text). The text has been changed to following:

*The increased concentrations of GEM may not only be caused by increased emission but part of the concentration increase could also be due to long-range transportation of GEM. Trajectory calculations of air mass transport on April 27 show downward mixing from higher elevations (Figure 3a), which could introduce air masses with higher GEM concentrations to our measurement site. However, at the same time we found upward fluxes of GEM, and in order to obtain an upward surface flux, the concentration in the snow must be higher than in the atmosphere.*

*The GEM emission on April 28 (event 2) was followed by an increase in GEM concentration on April 29. This occurred as the stability rapidly changed from stable ($z/L > 0$) to unstable ($z/L < 0$) conditions. The GEM concentration was relatively constant around 1 ng $m^{-3}$ on April 28 but increased threefold as the stratification changed from stable condition to unstable on April 29. According to trajectory calculations, this sudden increase was not caused by mixing from aloft (Figure 3b). We speculate that strongly stable conditions can result in GEM buildup directly above the surface, similar to $CO_2$ storage over forested sites (Yang et al., 2007). Surface emission of GEM into a relatively shallow layer of air will result in its higher concentration close to the ground. This buildup concentration would not be detected until the layer at the surface is mixed to a higher elevation when the stratification becomes unstable. On May 7, (Event 3) a change from stable to unstable conditions occurred simultaneously with an increase in concentration, which also partly could be explained by inversion of the surface layer as described above. The concentration increase was rapid, although not as large as the previous event (event 2), but the GEM emission in the days before event 3 were low and the stable conditions only lasted for a few hours (5-6 hours). Thus, we argue that the low GEM emissions lead to only a minor accumulation of GEM in the shallow surface layer before the surface layer was inverted. This concentration increase cannot be explained by a mixing from aloft as the trajectory calculations show a constant air mass transport pattern from May 3 to May 6 (Figures 3c and 3d), which should preclude such an event. There are other cases of stability change during our measurement period, but often the wind speed is higher, thus a shallow surface layer may perhaps not be formed. If a "build up" or "storage" effect exists, the flux measurements are also affected, and evaluation of flux data becomes even more complicated, thus, a more detailed study of the structure and dynamic of the Arctic atmospheric surface layers is needed.*

*This "shallow stable layer - inversion mechanism" is just a hypothesis, however, if this is a general pattern for very stable conditions, this can be an important effect, which needs to be considered in future measurements of Hg concentrations in the high Arctic. According to Osterwalder et al. (2016), GEM REA fluxes were significantly different under stable, unstable and neutral conditions over a snow-covered surface. In the present study, GEM was primarily emitted under neutral and slightly stable conditions, and fluxes close to zero were observed under unstable and neutral conditions. On the other hand, Osterwalder et al. (2016) observed emission during unstable conditions, a small deposition during stable conditions and deposition during neutral conditions. The differences in emission during certain stabilities can be explained by a non-Arctic location and a very different dynamic of GEM.*

**Minor Comment 1**. I checked with the Sommar et al. (2013) paper to see if their notation is consistent with that used for the equations (1) and (2) in the present work by Kamp et al. In Sommar et al. (2013) (their page 6, left column), the difference between the true and measured concentrations is denoted by the

capital C and the small letter c. It appears that the present paper does the same for the equation (2), but not for the equation (1). Also, it should be clearly stated that the updraft and downdraft concentrations referred to in the equation (1) and on the LHS of the equation (2) are the true ones, whereas those on the RHS of the equation (2) are the measured values that should be corrected for the zero-air dilution.

Respond: Notation for true and measured concentrations can be confused. "True" is added to the explanation on eq. 1:

$C_{up}$ and $C_{down}$ *are the true gas concentration*

The following line is added to the explanation on eq. 2:

$C_{up}$ and $C_{down}$ *are true corrected concentrations used in equation 1.*

**Minor Comment 2**. In Figures 9b and 10a, there are data points associated with the GEM fluxes in excess of 100 ng m-2 min-1 (at temperatures higher than -10 degrees Celsius and at wind speeds greater than 10 m s-1). Do they correspond to the "outlier" event on April 30 (page 7, last paragraph)? If so, they need to be annotated appropriately by using different colors and/or symbols.

Respond: We have changed the figures 9 and 10 so the special events are colored red.

**Minor Comment 3**. "Event 3" marked in figures 6-8 is not explicitly referred to in the text (section 3).

Respond: It is now referred to in the text.

**Technical suggestions**

P1, L14: … with only a few MINOR EPISODES OF NET depositional fluxes, FROM a maximum deposition of 8.1 ng m-2 min-1 TO a maximum emission of 179.2 ng m-2 min-1.

Respond: Changed to the suggested.

P2, L6: … between ozone and GEM concentrationS

Respond: Changed to the suggested.

P2, L8: mutual -> common

Respond: Changed to the suggested.

P3, L4: product -> consequence

Respond: Changed to the suggested.

P4, L28: than this threshold FOR AIR SAMPLES to be collected

Respond: Changed to the suggested.

P5, L1: Drop "are" at the end of the line.

Respond: Changed to the suggested.

P5, L9: Is the acronym "FPGA" defined? Or, is it obvious to most readers?

Respond: FPGA is obvious to readers working with similar data collecting systems. Left in to ensure reproducibility. Nothing is added.

P5, L29: IN CONTRAST TO THE GEM FLUX, the CO2 flux can be measured…

Respond: Changed to the suggested.

P6, L21: … a fixed dead band causing b to vary with …

Respond: Changed to the suggested.

P6, L24: "L" and "z" are defined already earlier in section 2.5.

Respond: Definition removed.

P6, L28-29: … if they fall OUTSIDE the stability range OF -1.5 < z/L < 1.5.

Respond: Changed to the suggested.

P7, L24-25: As expected, the large emission events were connected to increased wind speed and resultant increase in turbulent transport.

Respond: Changed to the suggested.

P8, L11: IN the present study, …

Respond: Changed to the suggested.

P8, L20: Is "the water phase" a terminology used in Steffen et al. (2015)? I would rather rephrase it to something else, e.g. "the liquid or liquid-like entities"

Respond: It is changed to the aqueous phase as used in the reference.

[revised manuscript text omitted]